

# Evaluating Simplified Methods for Liquefaction Assessment for Loss Estimation

Indranil Kongar[1], Tiziana Rossetto[1], Sonia Giovinazzi[2]

[1]Earthquake and People Interaction Centre (EPICentre), Department of Civil, Environmental and Geomatic Engineering, University College London, London, WC1E 6BT, United Kingdom

[2]Department of Civil and Natural Resources Engineering, University of Canterbury, Christchurch, 8140, New Zealand

*Correspondence to*: Indranil Kongar (ucfbiko@ucl.ac.uk)

**Abstract.** Currently, catastrophe models used by the insurance industry account for liquefaction simply by applying a factor to shaking-induced losses based on local liquefaction susceptibility so there is a need more sophisticated approach to incorporating the effects of liquefaction in loss models is needed. This study compares eleven unique models, each based on one of three principal simplified liquefaction assessment methodologies: liquefaction potential index (LPI) calculated from shear-wave velocity; the HAZUS software methodology; and a methodology created specifically to make use of USGS remote sensing data. Data from the September 2010 Darfield and February 2011 Christchurch earthquakes in New Zealand are used to compare observed liquefaction occurrences to predictions from these models using binary classification performance measures. The analysis shows that the best performing model is the LPI calculated using known shear-wave velocity profiles, although this data may not always be available to insurers. The next best model is also based on LPI, but uses shear-wave velocity profiles simulated from the combination of USGS $V_{S30}$ data and empirical functions that relate $V_{S30}$ to average shear-wave velocities at shallower depths. This model is useful for insurance since the input data is publicly available. This paper also considers two models (HAZUS and EPOLLS) for prediction of the scale of liquefaction in terms of permanent ground deformation but finds that both models perform poorly and thus potentially provide negligible additional value to loss estimation analysis outside of the regions for which they have been developed.

## 1 Introduction

The recent earthquakes in Haiti (2010), Canterbury, New Zealand (2010-11) and Tohoku, Japan (2011) highlighted the significance of liquefaction as a secondary hazard of seismic events and the significant damage that it can cause to buildings and infrastructure. However, the insurance sector was caught out by these events, with catastrophe models underestimating the extent and severity of liquefaction that occurred (Drayton and Verdon, 2013). A contributing factor to this is that the method used by some catastrophe models to account for liquefaction is based only on liquefaction susceptibility, a



qualitative parameter that considers only surficial geology characteristics. Furthermore, losses arising from liquefaction are predicted by adding an amplifier to losses predicted due to building damage caused by ground shaking (Drayton and Vernon 2013). There is a paucity of past event data on which to calibrate an amplifier and consequently, significant losses from liquefaction damage will only be predicted if significant losses are already predicted from ground shaking, whereas it is

known that liquefaction can be triggered at relatively low ground shaking intensities (Quigley et al., 2013).

Therefore there is scope within the insurance and risk management sectors to adopt more sophisticated approaches for predicting liquefaction for both future risk assessment and post-event rapid response analysis. It is also important to develop a better understanding of the correlation between liquefaction risk and physical damage of the built environment, similar to

the fragility functions that are used to predict damage associated with ground shaking. This is particularly the case for critical infrastructure systems since, whilst liquefaction is less likely than ground shaking to be responsible for major building failures (Bird and Bommer, 2004), it can have a major impact on lifelines such as roads, pipelines and buried cables. Loss of power and reduction in transport connectivity are major factors affecting the resilience of business organizations in response to earthquakes as they can delay the recommencement of normal operations. Evaluating the seismic performance of

infrastructure is therefore critical to understanding indirect economic losses caused by business interruption and to achieve this it is necessary to assess the liquefaction risk in addition to that posed by ground shaking.

Bird and Bommer (2004) surmised that there are three options that loss estimators can select to deal with ground failure hazards. They can either ignore them; or use a simplified approach; or conduct a detailed geotechnical assessment. The first

of these options will likely lead to underestimation of losses in earthquakes where liquefaction is a major hazard and lead to recurrence of the problems faced by insurers following the 2010-11 Canterbury earthquakes in particular. The last option, detailed assessment, is appropriate for single-site risk analysis but is impractical for insurance loss estimation purposes because: 1) insurers are unlikely to have access to much of the detailed geotechnical data required as inputs to these methods; 2) they may not have the in-house expertise to correctly apply such methods and engaging consultants may not be a

viable option; and 3) loss estimation studies are often conducted on a regional, national or supra-national scale for which detailed assessment would be too expensive and time-consuming.

There are three stages to predict the occurrence of liquefaction and its scale (Bird et al., 2006). First it is necessary to determine whether soils are susceptible to liquefaction. Liquefaction susceptibility is based solely on ground conditions with

no earthquake-specific information. This is often done qualitatively and currently this is also the full extent to which liquefaction risk is considered in some catastrophe models (Drayton and Verdon, 2013). The next step is to determine liquefaction triggering, which determines the likelihood of liquefaction for a given earthquake based on the susceptibility and other earthquake-specific parameters. Finally the scale of liquefaction can be predicted as a permanent ground



deformation (PGDf). Since current catastrophe modelling practice is to consider only the first stage, liquefaction susceptibility, this paper focuses primarily on the extension of this practice to include liquefaction triggering.

The models assessed in this paper have been selected because their input requirements are limited to data that are in the public domain or could be easily obtained without significant time or cost implications, arising for example from detailed site investigation. Furthermore, the models are appropriate for regional-scale analysis and although some engineering judgment is required in their application, they do not require specialist geotechnical expertise. In section 2, each of the models assessed in this paper are described and section 3 presents a summary of the liquefaction observations from the Canterbury earthquake sequence and the methodology used to test the predictive capabilities of the models against observations. The results and statistical analysis of the model assessment are presented in section 4, in relation to deterministic predictions, and in section 5, in relation to probabilistic predictions. Finally section 6 briefly considers the predictive capability of simplified models for quantifying PGDf.

## 2 Liquefaction assessment models

Nine liquefaction prediction models are tested in this paper, including three alternative implementations of the liquefaction potential index method proposed by Iwasaki et al. (1984); three versions of the liquefaction models included in the HAZUS$^{\text{®MH}}$ MR4 software (NIBS, 2003); and three distinct models proposed by Zhu et al. (2015). This section summarises how each of the models are applied to make site-specific liquefaction predictions. This paper presents a large number of acronyms and variables. For clear reference, Table 1 lists the acronyms used in this paper and Table 2 lists the variables used.

### 2.1 Liquefaction potential index

The most common approach used to predict liquefaction triggering is the factor of safety against liquefaction, *FS*, which is defined as the ratio of the cyclic resistance ratio, *CRR*, and the cyclic stress ratio, *CSR*, for a layer of soil at depth, *z* (Seed and Idriss, 1971). *CSR* can be expressed by:

$$CSR = 0.65\left(\frac{a_{\max}}{g}\right)\left(\frac{\sigma_v}{\sigma_v'}\right)r_d \tag{1}$$

where $a_{\max}$ is the peak horizontal ground acceleration; $g$ is the acceleration of gravity; $\sigma_v$ is the total overburden stress at depth $z$; $\sigma_v'$ is the effective overburden stress at depth $z$; and $r_d$ is a shear stress reduction coefficient given by:




$$r_d = 1 - 0.00765z, \text{ for } z < 9.2\text{m}$$
$$r_d = 1.174 - 0.0267z, \text{ for } z \geq 9.2\text{m}$$
(2)

$CRR$ is normally calculated from geotechnical parameters based on cone penetration test (CPT) or standard penetration test (SPT) results. However, Andrus and Stokoe (2000) propose an alternative method for calculating $CRR$ based on shear-wave velocity, $V_S$, where:

$$CRR = \left[ 0.022 \left( \frac{V_{S1}}{100} \right)^2 + 2.8 \left( \frac{1}{V_{S1}^* - V_{S1}} - \frac{1}{V_{S1}^*} \right) \right] \times MSF$$
(3)

where $V_{S1}$ is the stress-corrected shear wave velocity; $V_{S1}^*$ is the limiting upper value of $V_{S1}$ for cyclic liquefaction occurrence, which varies between 200-215m/s depending on the fines content of the soil; and $MSF$ is a magnitude scaling factor. $V_{S1}$ is given by:

$$V_{S1} = V_S \left( \frac{P_a}{\sigma_v'} \right)^{0.25}$$
(4)

where $P_a$ is a reference stress of 100kPa. The magnitude scaling factor is given by:

$$MSF = \left( \frac{M_W}{7.5} \right)^{-2.56}$$
(5)

15 where $M_w$ is the moment magnitude of the earthquake. Liquefaction is predicted to occur when $FS \leq 1$, and predicted not to occur when $FS > 1$. However Juang et al. (2005) found that Eq. (3) is conservative for calculating $CRR$, resulting in lower factors of safety and over-prediction of liquefaction occurrence. To correct for this, they propose a multiplication factor of 1.4 to obtain an unbiased estimate of the factor of safety, $FS^*$:

$$FS^* = 1.4 \times \frac{CRR}{CSR}$$
(6)

$FS^*$ is an indicator of potential liquefaction at a specific depth. However, Iwasaki et al. (1984) noted that damage to structures due to liquefaction was affected by the severity of liquefaction at ground level and so propose an extension to the



factor of safety method, the liquefaction potential index, *LPI*, which predicts the likelihood of liquefaction at surface-level by integrating a function of the factors of safety for each soil layer within the top 20m of soil. They calculate *LPI* as:

$$LPI = \int_0^{20} F^* \left(10 - 0.5z\right) dz \qquad (7)$$

where $F^* = 1 - FS^*$ for a single soil layer. The soil profile can be sub-divided into any number of layers (e.g. twenty 1m layers or forty 0.5m layers), depending on the resolution of data available. Using site data from a collection of nine Japanese earthquakes between 1891 and 1978, Iwasaki et al. (1984) calibrated the *LPI* model and determined guideline criteria for determining liquefaction risk. These criteria propose that liquefaction risk is very low for *LPI* = 0; low for 0 < *LPI* ≤ 5; high for 5 < *LPI* ≤ 15; and very high for *LPI* > 15.

One of the critical considerations for insurers is availability of model input data. For post-event analysis, ground accelerations may be available from USGS ShakeMaps (USGS, 2014a), but otherwise it would be necessary to apply engineering judgment in the selection of an appropriate ground motion prediction equation. The *LPI* model also requires water table depth and soil unit weights. If these are not known exactly, engineering judgment needs to be applied to estimate

these based on information in existing literature. Although the use of $V_S$ negates the requirement for ground investigation, $V_S$ data itself is not commonly in the public domain and may not necessarily be available across the entire study area, thus requiring geostatistical techniques to extrapolate. Consequently this method may only be applicable in a small number of study areas.

To extend the applicability of the *LPI* model, two approaches are proposed to approximate $V_S$ from more readily available data. The first approach uses $V_{S30}$, the average shear wave velocity across the top 30m of soil, as a constant proxy for $V_S$ for all soil layers. Global estimates for $V_{S30}$ at approximately 674m grid intervals are open-access from the web-based US Geological Survey Global $V_{S30}$ Map Server (USGS, 2013), so this is an appealing option for desktop assessment. The disadvantage of this approach is that the likelihood of liquefaction occurrence in the LPI method is controlled by the

presence of soil layers near the surface with low $V_S$. Furthermore there is a maximum value of $V_S$ at which liquefaction can occur. Hence the use of $V_{S30}$ as a proxy for all layers will result in an overestimation of $V_S$, *CRR* and $FS^*$ at layers closer to the surface, and therefore an underestimation of *LPI* and liquefaction risk.

The second approach proposes the use of the same $V_{S30}$ data but manipulates it to simulate a more realistic $V_S$ profile in
which velocities decrease towards the surface rather than being constant. Boore (2004) proposes empirical functions to extrapolate $V_{S30}$ values in situations where shear wave velocity data are only known up to shallower depths, based on observations from the United States and Japan. It is proposed to use these empirical functions in reverse – to back-calculate




shallower average shear wave velocities from $V_{S30}$ data from the USGS Global Server (USGS, 2013). For simplicity it is proposed to only use the empirical functions to calculate $V_{S10}$ (average shear wave velocity across top 10m) and $V_{S20}$ (average shear wave velocity across top 20m). The calculated value for $V_{S10}$ can then be used as a proxy for $V_S$ at all soil layers between 0-10m depth and both the $V_{S10}$ and $V_{S20}$ values can be used to determine an equivalent proxy for all soil layers

between 10-20m. From manipulation of the Boore (2004) empirical functions and the formula for calculating averaged shear wave velocities, the following equations determine the proxies to be used in the two depth ranges:

$$V_{S(0-10)} = 10^{\left(\frac{\log V_{S30} - 0.042062}{1.0292}\right)} \tag{8}$$

$$V_{S(10-20)} = \frac{1}{\dfrac{2}{10^{\left(\frac{\log V_{S30} - 0.025439}{1.0095}\right)}} - \dfrac{1}{V_{S(0-10)}}} \tag{9}$$

In this study, both of these approximations are adopted in addition to the use of known $V_S$ profiles, resulting in the assessment of three implementations of the *LPI* model.

## 2.2 HAZUS

HAZUS[®MH] MR4 (from here on referred to as HAZUS) is a loss estimation software package produced by the National Institute of Building Sciences (NIBS) and distributed by the Federal Emergency Management Agency (FEMA) in the United

States. The software accounts for the impacts of liquefaction and the Technical Manual (NIBS, 2003) describes the methodology used to evaluate the probability of liquefaction.

HAZUS divides the assessment area into six zones of liquefaction susceptibility, from none to very high. This can be done by either, interpreting surficial geology from a map and cross-referencing with the table published in the Manual, or by using

an existing liquefaction susceptibility map. Surface geology maps are generally not open-access or free to non-academic organizations and some basic geological knowledge is required to be able to cross-reference mapped information with the zones in the HAZUS table. Hence, the first approach may be problematic for insurers who do not have the requisite in-house expertise. Where liquefaction susceptibility maps are available, unless they use the same zonal definitions as HAZUS, it will be necessary to make assumptions on how zones translate between the third party map and the Manual.

For a given liquefaction susceptibility category, the probability of liquefaction occurrence is given by (NIBS, 2003):



$$P[Liq] = \frac{P[Liq \mid PGA = a]}{K_M K_W} P_{ml} \tag{10}$$

where $P[Liq \mid PGA=a]$ is the conditional probability of liquefaction occurrence for a given susceptibility zone at a specified level of peak horizontal ground motion, $a$; $K_M$ is the moment magnitude correction factor; $K_w$ is the ground water correction factor; and $P_{ml}$ is the proportion of map unit susceptible to liquefaction, which accounts for the real variation in susceptibility across similar geologic units. The conditional probability is zero for the susceptibility zone 'None' and for the other zones are given by linear functions of acceleration (distinct for each zone), which are not repeated here. The moment magnitude and ground water correction factors are given by:

$$K_M = 0.0027M_W{}^3 - 0.00267M_W{}^2 - 0.2055M_W + 2.9188 \tag{11}$$

$$K_W = 0.022d_w + 0.93 \tag{12}$$

where $d_w$ is the depth to ground water. The map unit factor is a constant for each susceptibility zone, with values of 0.25, 0.20, 0.10, 0.05, 0.02 and 0, going from 'Very high' to 'None'. In addition to the problems identified for determining liquefaction susceptibility, the HAZUS method also requires water table depth to be known or estimated and judgment on selection of appropriate ground motion prediction equation if ShakeMap or equivalent data is not available.

**2.3 Zhu et al. (2015)**

Zhu et al. (2015) propose empirical functions to predict liquefaction probability specifically for use in rapid response and loss estimation. They deliberately use predictor variables that are readily accessible, such as $V_{S30}$ and do not require any specialist knowledge to be applied. The functions have been developed using logistic regression on data from the earthquakes that occurred in Kobe, Japan on January 17[th] 1995 and in Christchurch, New Zealand on February 22[nd] 2011. The resulting functions have been tested on observations from the January 12[th] 2010 Haiti earthquake. For a given set of predictor variables, the probability of liquefaction is given by the function:

$$P[Liq] = \frac{1}{1 + e^{-X}} \tag{13}$$

where $X$ is a linear function of the predictor variables. Zhu et al. (2015) propose three linear models that are applicable to the Canterbury region and are adopted in this study: a specific local model for Christchurch; a regional model for use in coastal sedimentary basins (including Christchurch) and a global model that is applicable more generally. For the global model, the linear predictor function, $X_G$, is given by:



$$X_G = 24.1 + \ln PGA_{M,SM} + 0.355CTI - 4.784 \ln V_{S30} \qquad (14)$$

where, *CTI* is the compound topographic index, used as a proxy for saturation, and which can be obtained globally from the USGS Earth Explorer web service (USGS, 2014b); $V_{S30}$ is obtained from the USGS Global Server (USGS 2013); and

$PGA_{M,SM}$ is the product of the peak horizontal ground acceleration from ShakeMap estimates (USGS 2014a)  and a magnitude weighting factor, *MWF*, given by:

$$MWF = \frac{M_W^{2.56}}{10^{2.24}} \qquad (15)$$

For the regional model, the linear predictor function, $X_R$, is given by:

$$X_R = 15.83 + 1.443 \ln PGA_{M,SM} + 0.136CTI - 9.759ND - 2.764 \ln V_{S30} \qquad (16)$$

where additionally, *ND* is the distance to the coast, normalized by the size of the basin, i.e. the ratio between the distance to the coast and the distance between the coast and inland edge of the sedimentary basin (soil/rock boundary). The location of the inland edge can be estimated from a surface roughness calculation based on a digital elevation model (USGS, 2014b) or

by using $V_{S30}$ data such that the inland edge is assumed to be the boundary between NEHRP site classes C (soft rock) and D (stiff soil), (i.e. at $V_S$ = 360m/s). For the Christchurch-specific local model, the linear predictor function, $X_L$, is given by:

$$X_L = 0.316 + 1.225 \ln PGA_{M,SM} + 0.145CTI - 9.708ND \qquad (17)$$

For applicability within the insurance sector, this model presents an advantage over LPI and HAZUS since the only

parameter that requires engineering judgment is the selection of ground motion prediction equation if ShakeMap or equivalent data is not available.

## 3 Model test application

This section summarises the procedure for compatring the model predictions to observations from the Canterbury eartqhuake sequence. A brief description is provided of the liquefaction observation dataset and the additional datasets accessed in order

to provide the required inputs to the nine models. This is followed by a discussion on the conversion of quantitative model outputs to categorical liquefcation predictions and an explanantion of the test diagnostics used to assess model performance.



### 3.1 Liquefaction observations

The methods described in the previous section are tested for two case studies from the Canterbury earthquake sequence: the $M_W$ 7.1 Darfield earthquake on September 4th 2010 and the $M_W$ 6.2 Christchurch earthquake on February 22nd 2011, as identified in Figure 1. The corresponding peak horizontal ground acceleration contours for each earthquake are shown in

Figure 2.

Surface liquefaction observation data has been obtained from two sources: ground investigation data provided directly from Tonkin & Taylor, geotechnical consultants to the New Zealand Earthquake Commission (EQC) (van Ballegooy et al., 2014) and maps stored within the Canterbury Geotechnical Database (CGD, 2013a), an online repository of geotechnical data and

reports for the region set up by EQC for knowledge sharing after the earthquakes. The data provided by Tonkin & Taylor includes records from over 7,000 geotechnical investigation sites across Christchurch. After each earthquake, a land damage category is attributed to each site, representing a qualitative assessment of the scale of liquefaction observed. There are six land damage categories, but since this study only investigates liquefaction triggering, the categories are converted to a binary classification of liquefaction occurrence. This data is supplemented by the maps from the CGD which show the areal extent

of the same land damage categories. To ensure equivalence in the test, all models are applied to the same test area for each earthquake, which is the region for which the input data for all models is available. The test area is divided into a grid of 100m x 100m squares, generating 25,100 test sites. It is noted however that at some locations within Christchurch, no liquefaction observations are available so these sites are excluded from the subsequent analysis. As a result, the test area consists of 20,147 sites for the Darfield earthquake and 22,803 sites for the Christchurch earthquake. The observations from

the two events are shown in Figure 3.

### 3.2 Prediction model inputs

This study includes three implementations of the LPI model: 1) using known $V_S$ profiles (referred to as LPI1 in this paper); 2) using $V_{S30}$ as a proxy for $V_S$ (LPI2); and 3) using 'realistic' $V_S$ profiles simulated from $V_{S30}$ and the Boore (2004) functions (LPI3). The geotechnical investigation data provided by Tonkin & Taylor also includes values of *LPI* calculated at each site

from SPT data rather than $V_S$. Although this approach is not feasible for insurers, for reference its predictive power is also tested here. This implementation is referred to as LPIref. A water table depth of 2m has been assumed across Christchurch, reflecting the averages described by Giovinazzi et al. (2011) – 0-2m in the eastern suburbs and 2-3m in the western suburbs – and soil unit weights of 17kPa above the water table and 19.5kPa below the water table are assumed, as suggested by Wotherspoon et al. (2014). $V_{S30}$ data for LPI2 and LPI3 is taken from the USGS web server, with point estimates on an

approximately 674m grid.



Wood et al. (2011) have published $V_S$ profiles for 13 sites across Christchurch obtained using surface wave testing methods. In GIS, the profiles are converted to point data for each 1m depth increment from 0-20m, so that each point represents the $V_S$ at that site for a single soil layer and there are a total of 13 points for each soil layer. Ordinary kriging (with log transformation to ensure non-negativity) is applied to the points in each soil layer to create interpolated $V_S$ raster surfaces for

each layer. Whilst Andrus and Stokoe (2000) advise that the maximum $V_{S1}$ can range from 200-215m/s depending on fines content, subsequent work by Zhou and Chen (2007) indicates that the maximum $V_{S1}$ could range between 200-230m/s. In the absence of specific fines content data, a median value of 215m/s is assumed to be the maximum. In practice, a soil layer may have a value of $V_{S1}$ below this threshold but not be liquefiable because the soil is not predominantly clean sand. Because of the regional scale of this analysis though, site-specific soil profiles (as distinct from $V_S$ profile) are not taken into account in

determining whether a soil layer is liquefiable. Goda et al. (2011) suggest the use of 'typical' soil profiles to determine the liquefaction susceptibility of a soil layer at a regional scale. Borehole data at sites close to the 13 $V_S$ profile sites are available from the Canterbury Geotechnical Database (CGD, 2013c). These indicate that in the eastern suburbs of Christchurch, soil typically consists predominantly of clean sand to 20m depth, with some layers of silty sand. On the western side of Christchurch however there is an increasing mix of sand, silt and gravel in soil profiles, particularly at depths down to 10m.

Therefore it is possible, particularly in western suburbs, that the calculated $V_{S1}$ values may indicate liquefiable soil layers when they are in fact not, which would lead to overestimation of *LPI* and the extent of liquefaction.

For application of the HAZUS method, liquefaction susceptibility zones have to be identified to determine the values of model input parameters. In this paper liquefaction susceptibility zones are adopted from the liquefaction susceptibility map

available from the Canterbury Maps web-resource operated by Environment Canterbury Regional Council (ECan, 2014). From the map it is possible to identify four susceptibility zones: 'None', 'Low', 'Moderate' and 'High'. However, six susceptibility zones are defined by HAZUS (NIBS, 2003). Since the Canterbury zones cannot be sub-divided, it is necessary to map the Canterbury zones onto four of the HAZUS zones. In HAZ1 the zones are mapped simply by matching names; in HAZ2, the 'Low' and 'High' zones in Canterbury are mapped to the more extreme 'Very low' and 'Very high' zones in

HAZUS; and in HAZ3, the relevant input parameters for each zone are taken to be the average of those identified in HAZ1 and HAZ2. The mapping between susceptibility zones in each of the implementations described in Table 3. As with the LPI model, depth to water table is assumed to be 2m across Christchurch.

Three models proposed by Zhu et al. (2015) are tested in this paper: 1) the global model (referred to as ZHU1); 2) the

regional model (ZHU2); and 3) the local model (ZHU3). The *PGA* 'shakefields' from the Canterbury Geotechnical Database (CGD, 2013b) are used as equivalents to the USGS ShakeMap. *CTI* (USGS, 2014b), at approximately 1km resolution and $V_{S30}$ (USGS, 2013) are downloaded from the relevant USGS web resources. In total nine model implementations are being tested, based on three general approaches (see Table 4).





### 3.3 Site-specific prediction

When using probabilistic prediction frameworks, one can interpret the calculated probability as a regional parameter that describes the spatial extent of liquefaction rather than discrete site specific predictions, and indeed Zhu et al. (2015) specifically suggest that this is how their model should be interpreted. So for example, one would expect 30% of all sites

with a liquefaction probability of 0.3 to exhibit liquefaction and 50% of all sites with a liquefaction probability of 0.5, etc. However, when using liquefaction predictions as means to estimate structural damage over a wide area, it is useful to know not just the number of liquefied site but also where these sites are. This is particularly important for infrastructure systems since the complexity of these networks means that damage to two identical components can have significantly different impacts on overall systemic performance depending on the service area of each component and the level of redundancy built

in.

There are two ways to generate site specific predictions from probabilistic assessments. One approach is to group sites together based on their liquefaction probability, and then randomly assign liquefaction occurrence to sites within the group based on that probability, e.g. by sampling a uniformly distributed random variable. This method is good for ensuring that

the spatial extent of the site specific predictions reflect the probabilities, but since the locations are selected randomly it has limited value for comparison of predictions to real observations from past earthquakes. It can be more useful for generating site specific predictions for simulated earthquake scenarios.

Another method is to set a threshold value for liquefaction occurrence, so all sites with a probability above the threshold are

predicted to exhibit liquefaction and all sites with a probability below the threshold are predicted to not exhibit liquefaction. The disadvantage of this approach is that the resulting predictions may not reflect the original probabilities. For example if the designated threshold probability is 0.5 and all sites have a calculated probability greater than this (even if only marginally), then every site will be predicted to liquefy. Conversely if all sites have a probability below 0.5, then none of the sites will be predicted to liquefy. However since there is no random element to the determination of liquefaction occurrence,

the predictions are more definitive in spatial terms and hence more useful for the model testing in this study. Although not strictly a probabilistic framework, thresholds can also be used to assign liquefaction occurrence based on *LPI*, by determining a value above which liquefaction is assumed to occur.

For all of the methods however, the issue arises of what value the thresholds should take. No guidance is given for HAZUS,

whilst Zhu et al. (2015) propose a threshold of 0.3 to preserve spatial extent, although they also consider thresholds of 0.1 and 0.2. In their original study, Iwasaki et al. (1984) suggest critical values of *LPI* of 5 and 12 for liquefaction and lateral spreading respectively. However other localized studies where the *LPI* method has been applied have found alternative criteria that provide a better fit for observed data as summarized by Maurer et al. (2014). Since there is uncertainty in the





selection of threshold values, this study tests a range of values for each model. Both the observation and prediction datasets are binary classifications, so standard binary classification measures based on 2 x 2 contingency tables are used to test performance.

### 3.4 Test diagnostics

Comparison of binary classification predictions with observations is made by summarizing data into 2 x 2 contingency tables for each model. The contingency table identifies the true positives (*TP*), true negatives (*TN*), false positives (*FP*, Type I error) and false negatives (*FN*, Type II error). A good predictive model would predict both positive (occurrence of liquefaction) and negative (non-occurrence of liquefaction) results well. Diagnostic scores for each model can be calculated based on different combinations and functions of the data in the contingency tables. The true positive rate (*TPR* or

sensitivity) is the ratio of true positive predictions to observed positives. The true negative rate (*TNR* or specificity) is the ratio of true negative predictions to observed negatives. The false positive rate (*FPR* or fall-out) is the ratio of false positive predictions to true negatives. The best model would have a high *TPR* and *TNR* ($> 0.5$) and low *FPR* ($< 0.5$).

The results presented in a contingency table and associated diagnostic scores assume a single initial threshold value.

However, further statistical analysis is undertaken to optimize the thresholds in accordance with the observed data. For a single model, at a specified threshold, the Receiver Operating Characteristic (ROC) is a graphical plot of *TPR* against *FPR*. The line representing *TPR* = *FPR* is equivalent to random guessing (known as the chance or no-discrimination line). A good model has a ROC above and to the left of the chance line, with perfect classification occurring at (0,1). The diagnostic scores for each model are re-calculated with different thresholds and the resulting ROC values are plotted as a curve for the model.

Since better models have points towards the top left of the plot, the area under the ROC curve, *AUC*, is a generalized measure of model quality that assumes no specific threshold. Since the diagonal of the plot is equivalent to random guessing, *AUC* = 0.5 suggests a model has no value, while *AUC* = 1 is a perfect model. For a single point on the ROC curve, Youden's *J*-statistic is the height between the point and the chance line. The point along the curve which maximizes the *J*-statistic represents the *TPR* and *FPR* values obtained from the optimum threshold for that model.

As well as comparing the performance of simplified models to each other, it is also useful to measure the absolute quality of each model. Simply counting the proportion of correct predictions does not adequately measure model performance since it does not take into the account the proportion of positive and negative observations, e.g. a negatively biased model will result in a high proportion of correct predictions if the majority of observations are negative. The Matthews correlation coefficient,

*MCC*, is more useful for cases where there is a large difference in the number of positive and negative observations (Matthews, 1975). It is proportional to the chi-squared statistic for a 2 x 2 contingency table and its interpretation is similar





to Pearson's correlation coefficient, so it can be treated as a measure of the goodness-of-fit of a binary classification model (Powers, 2011). From contingency table data, *MCC* is given by:

$$MCC = \frac{TP \times TN - FP \times FN}{\sqrt{(TP+FP)(TP+FN)(TN+FP)(TN+FN)}}$$

(18)

## 5   4 Results

An initial set of results using 5 as a threshold value for the LPI models, 0.3 as a threshold for the ZHU models and 0.5 as a threshold value for the HAZUS models is shown in Table 5, alongside the corresponding diagnostic scores.

10   The LPI1 and LPIref models are the only models that meet the criteria of having *TPR* and *TNR* > 0.5 and *FPR* < 0.5, with the LPI1 model performing better despite being based on $V_S$ rather than ground investigation data. Table 3 shows that all HAZUS models are very good at predicting non-occurrence of liquefaction. However, this is only due to the fact that they are predicting no liquefaction all the time, and so their ability to predict the occurrence of liquefaction is extremely poor. The high *TNR* but relatively low *TPR* of the three ZHU models indicate that they all show a bias towards the prediction of non-occurrence of liquefaction. The difference between *TPR* and *TNR* is indicative of the level of bias in the model and this regard ZHU2, the regional model shows less bias than in ZHU1, the global model, as would be expected. The bias in the ZHU2 and ZHU3 models is approximately similar although the predictive power of ZHU2 is slightly better.

The LPI2 model, using $V_{S30}$ as a proxy, also shows a very strong bias towards predicting non-occurrence, which is expected since $V_{S30}$ generally provides an overestimate of $V_S$ for soil layers at shallow depth. At sites where the soil profile of the top 20   30m is characterized by some liquefiable layers at shallow depth with underlying rock or very stiff soil (e.g. in western and central areas close to the inland edge of the sedimentary basin), $V_{S30}$ will be high. Hence, this leads to false classification of shallow layers as non-liquefiable. The LPI3 model with simulated $V_S$ profiles exhibits good performance in the prediction of non-occurrence of liquefaction but only correctly predicts half of the positive liquefaction observations, indicating bias towards negative predictions. Although the $V_S$ profiles generated through this approach are more realistic than using a 25   constant $V_{S30}$ value, the $V_S$ at each layer is related to $V_{S30}$. Therefore, at sites characterized by a high $V_{S30}$ value with low $V_S$ values at shallow depths, even using Eq. (8) and Eq. (9) may not predict sufficiently low values of $V_{S1}$ to classify the shallow layers as liquefiable. Another factor in the LPI models is the use of the bias-correction factor proposed by Juang et al. (2005). Whilst this correction factor is appropriate when actual $V_S$ profiles are used, as in LPI1, it may not be appropriate for LPI2 and LPI3 where non-conservative proxies for $V_S$ are used and the resulting misclassification of liquefiable soil layers 30   balances the conservativeness of the Andrus and Stokoe (2000) *CRR* model. The sensitivity of the models to the correction





factor is tested by reproducing the contingency tables for LPI2 and LPI3 with the same threshold values but ignoring the correction factor for *FS*. These models are referred to as LPI2b and LPI3b and the new contingency table analysis is presented in Table 6.

These results show that not using the bias correction makes little difference to the performance of LPI2, as LPI2b still exhibits an extremely strong bias towards predicting non-occurrence of liquefaction. For LPI3 however, the difference is more significant. Without the correction factor, LPI3b meets the criteria of having both *TPR* and *TNR* values greater than 0.5. The bias towards the prediction of non-occurrence of liquefaction is still present although greatly reduced. However, the improved capability of predicting the occurrence of liquefaction is offset by a reduction in the capability of predicting non-
occurrence of liquefaction. The results in Table 3 and Table 4 demonstrate the performance of each model with a single initial threshold value. ROC analysis is used to optimize the thresholds and curves for the eleven simplified models and reference model are generated using the ROCR package in R (Sing et al., 2005), as shown in Figure 4. For this study, the threshold for the LPI models is assumed to be a whole number, while for the HAZ and ZHU models, the threshold is assumed to be a multiple of 0.05 subject to a minimum value of 0.1, which is the minimum tested by the Zhu et al. (2014).
The *AUC* values, maximum *J*-statistics, optimum thresholds and corresponding *TPR* and *TNR* values for all models are shown in Table 7.

With optimized thresholds all the LPI models, except LPI2 and all the ZHU models meet the *TPR* and *TNR* criteria (>0.5). All HAZ models and both versions of LPI2 have *AUC* values closer to the 'no value' criterion, suggesting that the problems
with these models lie not just with threshold selection, but more fundamentally with their composition and/or relevance to the case study being tested (noting that the HAZ models have been developed for analysis in the United States). The reason these are to the left of the chance line is because they are predicting non-occurrence of liquefaction at nearly every site and hence they are guaranteed a low *FPR* value. LPI1 is the best performing model according to both of the ROC diagnostics and although the optimum threshold value of 7 is higher than proposed by Iwasaki et al. (1984), it is within the range for
marginal liquefaction – 4 to 8 – proposed by Maurer et al. (2014) and so may be considered plausible. The two versions of the LPI3 model perform similarly and have reasonable diagnostic scores but LPI3b, without the correction factor, produces a more plausible optimum threshold value of 4. It is noted however that although the optimum threshold for LPI3 is 1, the *TPR* and *TNR* criteria are met with a threshold of 4 but with a lower model performance and greater negative prediction bias (*J*-statistic = 0.403, *TPR* = 0.523, *TNR* = 0.880).

The ZHU1 and ZHU2 models perform reasonably with *AUC* values and *J*-statistics slightly lower than the LPI3 models, but the optimum thresholds are at the minimum of the tested range, confirming the degree to which these models under-predict liquefaction occurrence. The ZHU2 model also meets the *TPR* and *TNR* criteria with a threshold value of 0.2, albeit with a greater prediction bias (*J*-statistic = 0.370, *TPR* = 0.555, *TNR* = 0.815). The ZHU3 model, despite being specific to





Christchurch, does not perform as well as ZHU1 or ZHU2. The reason for this anomaly may be because the ZHU models were calibrated to preserve the extent of liquefaction rather than to make site-specific predictions.

The absolute quality of models is tested by calculating *MCC*. In the preceding analysis, the best performing model is LPI1 and this has a value of *MCC* = 0.48. The correlation is only moderate, but nevertheless indicates that the model is better than random guessing. As part of a rapid assessment or desktop study for insurance purposes, this may be sufficient. LPI3 and LPI3b have *MCC* = 0.339 and 0.348 respectively, whilst LPIref has *MCC* = 0.29.

## 5 Probability of liquefaction

The threshold-based approach to liquefaction occurrence identification using the LPI models, provides a deterministic prediction. This may be considered sufficient for the simplified regional-scale analyses conducted for catastrophe modelling and loss estimation. However, a modeller may also want to establish a probabilistic view of liquefaction risk by relating values of *LPI* to probability of liquefaction occurrence. Since the occurrence of liquefaction at a site is a binary classification variable, it can be modelled by a Bernoulli distribution with probability of liquefaction, *p*, which depends on the value of *LPI*. With data from past earthquakes, functions relating *p* to *LPI* can be derived using a generalized linear model with probit link function. The probability of liquefaction occurring given a particular value of *LPI*, $\lambda$, is given by:

$$p_{LPI=\lambda} = P\big[Liq \,|\, LPI = \lambda\big] = \Phi\big(Y^*\big) \tag{19}$$

where $\Phi$ is cumulative normal probability distribution function and $Y^*$ is the probit link function given by:

$$Y^* = \beta_0 + \beta_1 \lambda \tag{20}$$

The link function is a linear model with *LPI* as a predictor variable and is derived from the individual site observations. Figure 5 displays the relationships between liquefaction probability and *LPI* fit by this method for the two best performing LPI models, LPI1 and LPI3b, including 95% confidence intervals. The relationships are accompanied by plots of the observed liquefaction rates, aggregated at each value of *LPI*. The plot for model LPI3b shows greater scatter of observed rates around the fit line than the plot for model LPI1, although in both cases the confidence interval is very narrow, which is a reflection of the large sample size. The confidence interval for LPI1 (±0.0014) is slightly narrower that the confidence interval for LPI3b (±0.0021), indicating that LPI1 is the better predictor of liquefaction probability, just as it is better at predicting liquefaction occurrence by *LPI* threshold. It is noted in both models though that the observed rates that are furthest away from the fit line are mostly those based on smaller sample sizes (arbitrarily defined here as 100) and which therefore



have less influence on the regression – since the use of individual site observations implicitly gives more weight to observations in the region of *LPI* values for which sample sizes are larger. Furthermore, the observed rates are themselves more unreliable for smaller sample sizes. For example, for model LPI1, observations based on more than 100 samples have an average margin of error of 0.05 whereas the average margin of error for smaller samples is 0.19. For model LPI3b, observations based on more than 100 samples have an average margin of error of 0.05 and this increases to 0.31 when considering the observations based on smaller samples.

The Hosmer-Lemeshow test (Hosmer and Lemeshow, 1980) is a commonly used procedure for assessing the goodness of fit of a generalized linear model when the outcome is a binary classification. However, Paul et al. (2013) show that the test is biased with respect to large sample sizes, with even small departures from the proposed model being classified as significant and consequently recommend that the test is not used for sample sizes above 25,000. Pseudo-$R^2$ metrics are also commonly used to test model performance (Smith and McKenna, 2013), but these compare the proposed model to a null intercept-only model rather than comparing the model predictions to observations. Although the purpose of the analysis in this section is to relate *LPI* to liquefaction probabilistically, contingency table analysis with a threshold probability to determine liquefaction occurrence remains an appropriate technique to test the fit of the model (Steyerberg et al., 2010). Assuming a threshold probability of 0.5, Table 8 presents summary statistics from the contingency table analysis of each model and also the coefficients of the corresponding probit link function.

Both models have values of *TPR* and *TNR* above 0.5 and the values are of a similar order to those obtained in Table 5 and Table 6 for the same models but using a value of *LPI* as a threshold. In particular however, values of *TNR* are now higher, which indicates that the probabilistic *LPI* model is better at predicting non-occurrence of liquefaction. This is important as over 80% of observations in this study are negative. The difference in values of *AUC* between Table 7 and Table 8 are negligible but the *J*-statistic for LPI3b with a threshold probability of 0.5 is considerably higher than the *J*-statistic that the optimal threshold found for LPI3b in Table 7. This suggests that LPI3b is best implemented as a probabilistic model for liquefaction occurrence. Overall these statistics indicate that both of the probabilistic *LPI* models proposed are good fits to the observed data.

## 6 Permanent ground deformation

The preceding sections have analysed methods for predicting liquefaction triggering, but for assessing the fragility of structures and infrastructure, it is more informative to be able to predict the scale of liquefaction, in terms of the permanent ground deformation (PGDf). In fact, fragility functions for liquefaction-induced damage are commonly expressed in these terms (Pitilakis et al., 2014). A summary of the available approaches for quantifying PGDf is provided by Bird et al. (2006), which also compares approaches for lateral movement, settlement and combined movement (volumetric strain). The majority



of these approaches require detailed geotechnical data as inputs, (e.g. median particle size, fines content). The likelihood that insurers possess or are able to acquire such data is low, which means that these approaches are not suitable for regional-scale rapid assessment. The lack of simplified models is not surprising given the small number of models that exist for liquefaction triggering assessment and that by definition measuring the scale of liquefaction is more complex. From the available models

in the literature, there are three that can be applied without the need for detailed geotechnical data: the EPOLLS regional model for lateral movement (Rauch and Martin, 2000) and the HAZUS models for lateral movement and vertical settlement (NIBS, 2003). To demonstrate the challenge faced by insurers looking to improve their liquefaction prediction capability, these models are compared to PGDf observations from the Darfield and Christchurch earthquakes. It should be noted that the HAZUS model has been developed specifically for the United States and the empirical data used to develop its constituent

parts comes mainly from California and Japan. The EPOLLS model is based on empirical data from the United States, Japan, Costa Rica and the Philippines.

## 6.1 Vertical settlement

A time series of LiDAR surface data for Christchurch has been produced from aerial surveys over the city, initially prior to the earthquake sequence in 2003, and subsequently repeated after the Darfield and Christchurch earthquakes. The surveys

are obtained from the Canterbury Geotechnical Database (2012a). The LiDAR surveys recorded the surface elevation as a raster at 5m-cell resolution. The difference between the post-Darfield earthquake survey and the 2003 survey represents the vertical movement due to the Darfield earthquake. Similarly the difference between the post-Christchurch earthquake and the post-Darfield earthquake surveys represents the movement due to the Christchurch earthquake. In addition to liquefaction, elevation changes recorded by LiDAR can also be caused by tectonic uplift. Therefore, to evaluate the vertical movement

due to liquefaction effects only, $PGDf_V$, the differences between LiDAR surveys have been corrected to remove the effect of the tectonic movement. Tectonic uplift maps have been acquired from the Canterbury Geotechnical Database (2013d). The only simplified method for calculating vertical settlement is from HAZUS (NIBS, 2003) in which the settlement is the product of the probability of liquefaction, as in Eq. (10), and the expected settlement amplitude, which varies according to liquefaction susceptibility zone, as described in Table 9.

The HAZUS model is applied with each of the three implementations used for predicting liquefaction probability in the liquefaction triggering analysis. Summary statistics of the $PGDf_V$ predictions from each implementation are presented in Table 10. This shows that the HAZUS model significantly underestimates the scale of liquefaction, regardless of how liquefaction susceptibility zones are mapped between the Canterbury and HAZUS classifications. The residuals have a

negative mean in each implementation indicating an underestimations bias. Furthermore, the maximum value predicted by HAZ1 and HAZ3 is smaller than the observed lower quartile. The coefficient of determination is also extremely low in each case, implying that there is little or no value in the predictions. It is important to note that there is a measurement error in the



LiDAR data itself of up to 150mm, as well as a uniform probability prediction interval around the HAZUS prediction. However, even when using the upper bound of the HAZUS prediction (two times the mean), only around 50% of predictions fall within the observation error range. These results suggest that the HAZUS model for predicting vertical settlement is not suitable for application in Christchurch.

## 6.2 Lateral spread

The LiDAR surveys for Christchurch also record the locations of reference points within a horizontal plane and the differences between these data have been used to generate maps identifying the lateral displacements caused by each earthquake on a grid of points at 56m intervals. Similarly to the elevation data, the lateral displacements have to be corrected for tectonic movements, although in this case the corrected maps have been obtained directly from the Canterbury Geotechnical Database (2012b).

The HAZUS model (NIBS 2003) for predicting ground deformation due to lateral spread is given by:

$$PGDf_H = K_\Delta \times E\left[ PGD \,|\, \left( PGA \,/\, PL_{SC} \right) = a \right] \qquad (21)$$

where, $K_\Delta$ is a displacement correction factor, which is a cubic function of earthquake magnitude, and the term on the right hand side is the expected ground deformation for a given liquefaction susceptibility zone, which is a function of the normalized peak ground acceleration (observed PGA divided by liquefaction triggering threshold PGA for that zone). The formulae for calculating these terms are not repeated here but can be found in the HAZUS manual (NIBS, 2003).

The EPOLLS suite of models for lateral spread (Rauch and Martin, 2000) includes proposed relationships for predicting ground deformation at a regional scale (least complex), at site specific scale without detailed geotechnical data and at site specific scale with detailed geotechnical data (most complex). In the regional EPOLLS model, $PGDf_H$ is given by:

$$PGDf_H = \left( 0.613 M_W - 0.0139 R_f - 2.42 PGA - 0.01147 T_d - 2.21 \right)^2 + 0.149 \qquad (22)$$

where, $R_f$ is the shortest horizontal distance to the surface projection of the fault rupture, and $T_d$ is the duration of ground motion between the first and last occurrence of accelerations $\geq 0.05g$ at each site. To calculate duration, there are 19 strong-motion accelerograph stations in Christchurch that record ground motions at 0.02s intervals. The records from each station for both earthquakes are available from the GeoNet website (GNS Science, 2014). $T_d$ is calculated at each station and then the value at intermediate sites is interpolated by ordinary kriging. Summary statistics of the predictions from the regional



EPOLLS and HAZUS models are presented in Table 11. The statistics show that none of the models predict PGDf$_H$ well. The EPOLLS model overestimates the scale of liquefaction, while the HAZUS models each show an underestimation bias. The mean residuals and root mean square error (RMSE) are higher for the EPOLLS model, suggesting that the HAZUS models perform slightly better, but this is of little significance since the coefficients of determination of the HAZUS models

are all extremely low. A mitigating factor is that the LiDAR data has a very large error – up to 0.5m – in the horizontal plane. Taking this into account, over 90% of HAZUS predictions are within the observation error range, although this needs to be interpreted in the context of the mean observed *PGDf$_H$* being 0.269m. Since the HAZUS model underestimates *PGDf$_H$*, and *PGDf$_H$* cannot be negative, the fact that so many predictions are within this error range is more a reflection of the size of the error relative to the values being observed. Consequently the statistics in Table 11 are more informative and these show

that the simplified models all perform poorly.

**7 Conclusions**

This study compares a range of simplified desktop liquefaction assessment methods that may be suitable for insurance sector where data availability and resources are key constraints. It finds that the liquefaction potential index, when calculated using shear-wave velocity profiles (LPI1) is the best performing model in terms of its ability to correctly predict liquefaction

occurrence both positively and negatively. Shear-wave velocity profiles are not always available to practitioners and it is notable therefore that the analysis shows that the next best performing model is the liquefaction potential index calculated with shear-wave velocity profiles simulated from USGS $V_{S30}$ data (LPI3b). Since it is based on USGS data, which is publicly accessible online, this method is particularly attractive to those undertaking rapid and/or regional scale desktop assessments.

The HAZUS methodology for estimating liquefaction probabilities performs poorly irrespective of triggering threshold. This is significant since HAZUS methodologies (not only in respect to liquefaction) are often used as a default model outside of the US when no specific local (or regional) model is available. Models proposed by Zhu et al. (2015) perform well and since they are also based on publicly accessible data, represent another viable option for desktop assessment. The only issue with these models is that they perform optimally with a low threshold probability of 0.1, which may lead to over-prediction of

liquefaction when applied to other locations.

As an extension of the liquefaction triggering analysis, this study also uses the observations to relate *LPI* to liquefaction probability for the two best-performing models. In the case of LPI3b, the model performance (as measured by Youden's *J*-statistic) actually improves significantly when employed with a threshold based on corresponding probability rather than

based directly on *LPI*. The final stage of liquefaction assessment is to measure the scale of liquefaction as PGDf. This study only briefly considers this aspect but shows that existing simplified models perform extremely poorly. Existing models show very low correlation with observations and strong prediction bias – underestimation in the case of HAZUS and





overestimation in the case of regional EPOLLS. Based on this analysis the predictions from these simplified models are highly uncertain and it is questionable whether they genuinely add any value to loss estimation analysis outside of the regions for which they have been developed.

**Competing interests**

The authors declare that they have no conflict of interest.

**Acknowledgments**

The authors are grateful to Sjoerd van Ballegooy of Tonkin and Taylor for providing data on observed land damage categories and liquefaction metrics. Funding for this research project has been provided by the UK Engineering and Physical Sciences Research Council and the Willis Research Network, through the Urban Sustainability and Resilience Doctoral
Training School at University College London.

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





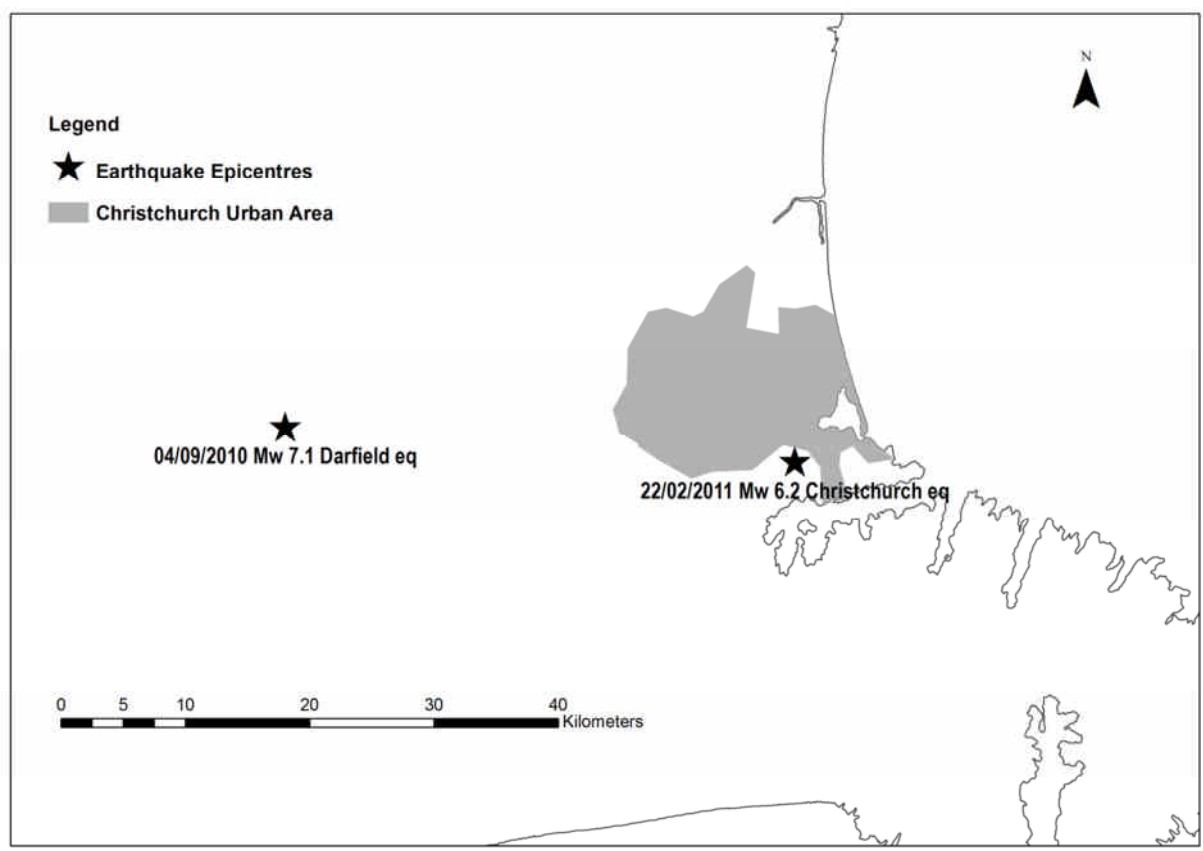

**Figure 1 – Location of epicentres of the Darfield and Christchurch earthquakes in relation to the Christchurch urban area and central business district.**





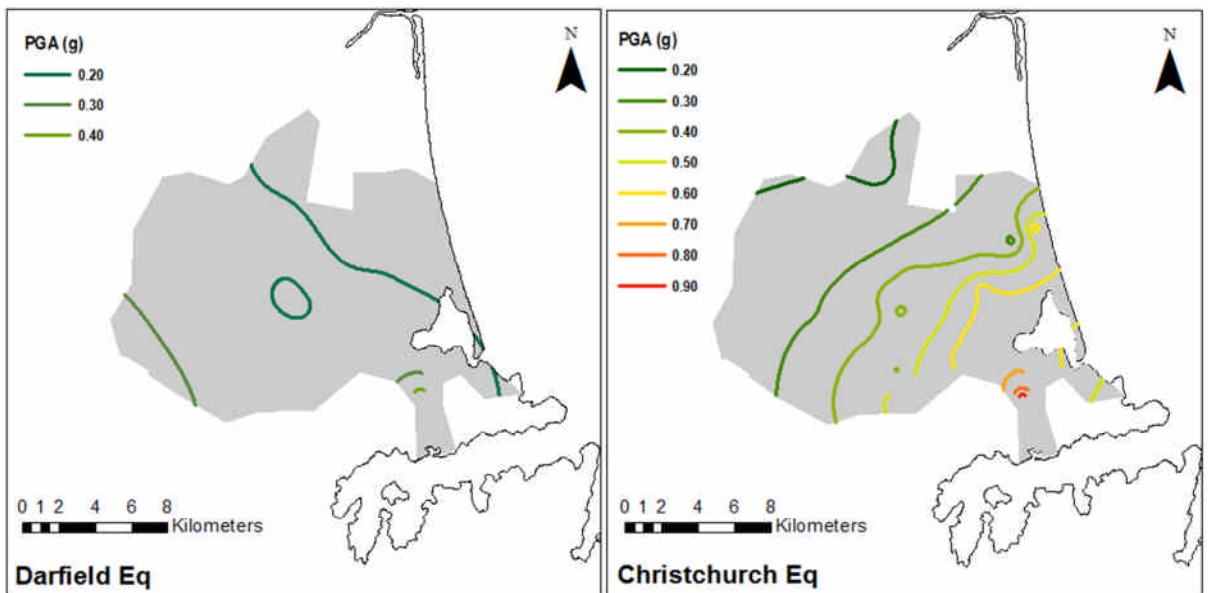

Figure 2 – Contours of peak horizontal ground acceleration for the Darfield and Christchurch earthquakes

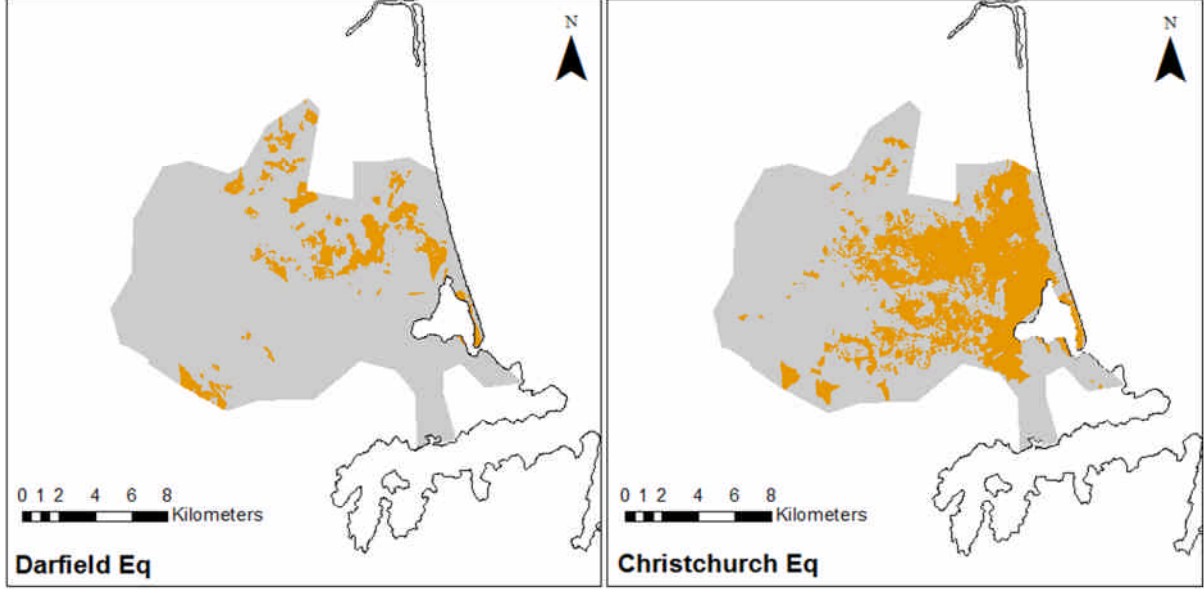

5    Figure 3 – Location of surface liquefaction observations (brown) in Christchurch and surrounding areas due to the Darfield and Christchurch earthquakes, based on data provided by Tonkin and Taylor and published within the Canterbury Geotechnical Database (CGD 2013a).





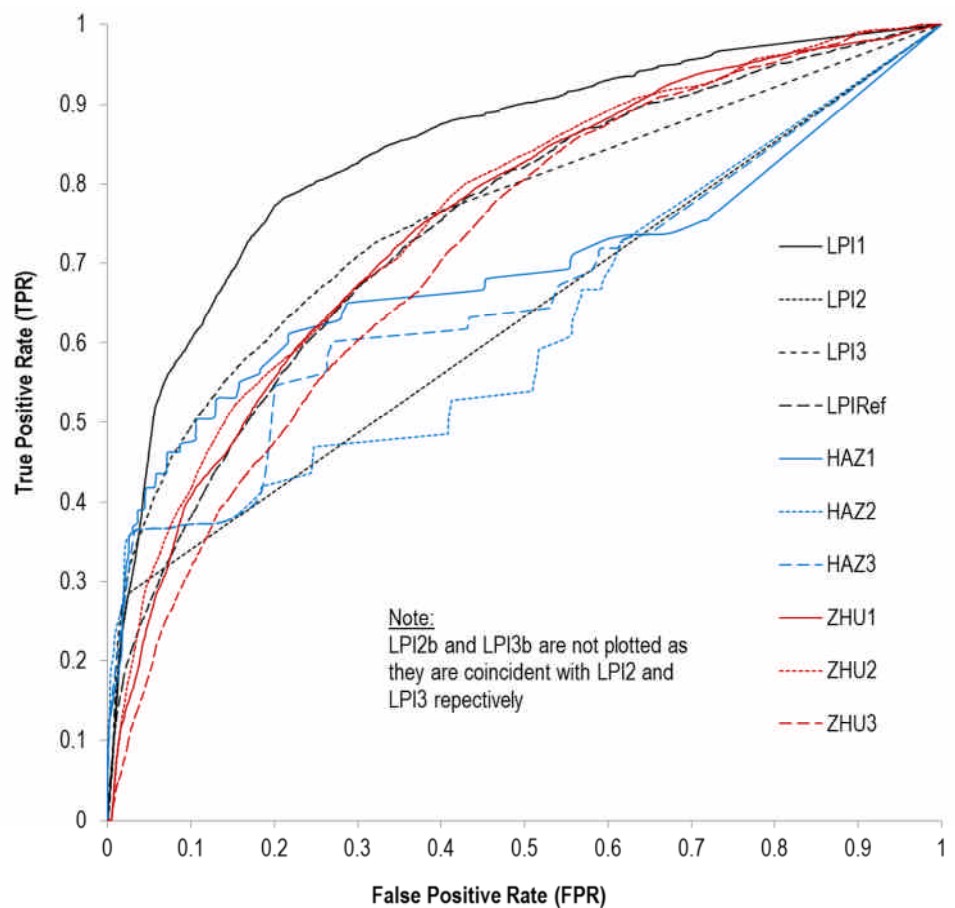

**Figure 4 – Receiver Operating Characteristic (ROC) curves for prediction models**




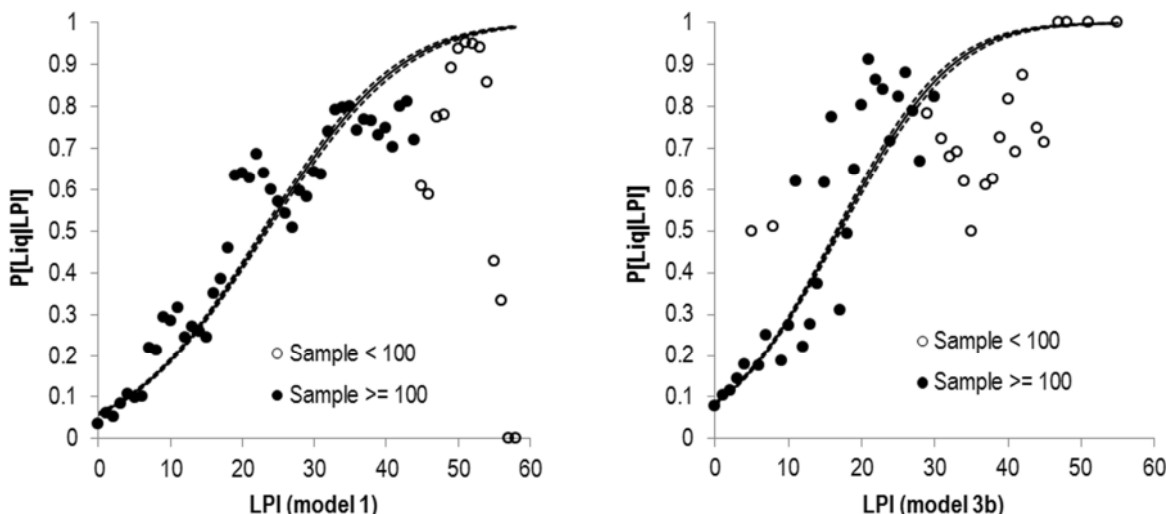

**Figure 5 – Plots of liquefaction probability against LPI derived from site specific observations by generalized linear model with probit link function for two best performing LPI models. Plots also display the observed liquefaction rates at each LPI value and classified by sample size**





**Table 1 – Reference list of acronyms used in this paper**

| Acronym | Description |
| --- | --- |
| AOC | Area under ROC curve |
| CGD | Canterbury Geotechnical Database |
| CPT | Cone penetration test |
| CRR | Cyclic resistance ratio |
| CSR | Cyclic stress ratio |
| CTI | Compund topographic index |
| EQC | Earthquake Commission |
| FN | False negative model predictions (no.) |
| FP | False positive model predictions (no.) |
| FPR | False positive rate ( = FP / Observed negatives) |
| FS | Factor of safety against liquefaction |
| LPI | Liquefaction potential index |
| MCC | Matthew's correlation coefficient |
| MSF | Magnitude scaling factor |
| MWF | Magnitude weighting factor |
| NEHRP | National Earthquake Hazards Reduction Program |
| PGA | Peak ground acceleration |
| PGDf | Permanent ground deformation |
| PGDf$_H$ | Horizontal permanent ground deformation |
| PGDf$_V$ | Vertical permanent ground deformation |
| RMSE | Root mean square error |
| ROC | Receiver operating characteristic |
| SPT | Standard penetration test |
| TN | True negative model predictions (no.) |
| TNR | True negative rate ( = TN / Observed negatives) |
| TP | True positive model predictions (no.) |
| TPR | True positive rate ( = TP / Observed positives) |
| USGS | United States Geological Survey |





**Table 2 – Reference list of variables used in this paper**

| Variable | Description | Units |
|---|---|---|
| $a_{max}$ | Peak horizontal ground acceleration | m/s$^2$ |
| $CRR$ | Cyclic resistance ratio | - |
| $CSR$ | Cyclic stress ratio | - |
| $CTI$ | Compund topographic index | - |
| $FS$ | Factor of safety against liquefaction | - |
| $K_M$ | HAZUS moment magnitude correction factor | - |
| $K_W$ | HAZUS ground water correction factor | - |
| $K_\Delta$ | Displacdement correction factor | - |
| $LPI$ | Liquefaction potential index | - |
| $MSF$ | Magnitude scaling factor | - |
| $M_W$ | Moment magnitude | - |
| $MWF$ | Magnitude weighting factor | - |
| $ND$ | Normalised distance to coast (Zhu et al., 2015) | - |
| $PGA$ | Peak horizontal ground acceleration (non USGS) | g |
| $PGA_{M,SM}$ | Peak horizontal ground acceleration (from USGS ShakeMap) | g |
| $PGD \mid (PGA / PLSC)$ | HAZUS expected $PGD_f$ for a given liquefcation susceptibility zone | m |
| $PGDf$ | Permanent ground deformation | m |
| $PGDf_H$ | Horizontal permanent ground deformation | m |
| $PGDf_V$ | Vertical permanent ground deformation | m |
| $P_{ml}$ | HAZUS proportion of map unit susceptible to liquefaction | - |
| $r_d$ | Shear stress reduction coefficient | - |
| $R_f$ | Horizontal distance to surface projection of fault rupture | km |
| $T_d$ | Duration between first and last occurrence of $PGA \geq 0.05g$ | s |
| $V_S$ | Shear wave vlocity | m/s |
| $V_{S1}$ | Stress-corrected shear wave velocity | m/s |
| $V^*_{S1}$ | Limiting upper value of $V_{S1}$ for cyclic liquefaction occurrence | m/s |
| $V_{S(0-10)}$ | Average shear wave velocity in top 10m | m/s |





| | | | |
|---|---|---|---|
| $V_{S(10\text{-}20)}$ | Average shear wave velocity between 10m and 20m | | m/s |
| $V_{S30}$ | Average shear wave velocity in top 30m | | m/s |
| $z$ | Depth | | m |
| $\sigma_v$ | Total overburden stress | | kPa |
| $\sigma_v'$ | Effective overburden stress | | kPa |

**Table 3 – Conversion between Canterbury and HAZUS liquefaction susceptibility zones for three implementations of HAZUS methodology**

| Canterbury liquefaction susceptibility zones | Equivalent HAZUS liquefaction susceptibility zone | | |
|---|---|---|---|
| | HAZ1 | HAZ2 | HAZ3 |
| None | None | None | None |
| Low | Low | Very low | Average low and very low |
| Moderate | Moderate | Moderate | Moderate |
| High | High | Very high | Average high and very high |

**Table 4 – Liquefaction prediction models tested in this paper**

| Model | Description |
|---|---|
| LPI1 | LPI with known $V_S$ profiles |
| LPI2 | LPI with $V_{S30}$ as $V_S$ proxy |
| LPI3 | LPI with simulated $V_S$ profiles |
| LPIref | LPI calculated from SPT results |
| HAZ1 | HAZUS with 'direct' conversion of susceptibility zones |
| HAZ2 | HAZUS with 'extreme' susceptibility zones |
| HAZ3 | HAZUS with 'average' conversion of susceptibility |
| ZHU1 | Global model by Zhu et al. (2015) |
| ZHU2 | Regional model by Zhu et al. (2015) |
| ZHU3 | Local model by Zhu et al. (2015) |





**Table 5 – Summary of contingency table data and diagnostic scores for all models using initial threshold estimates**

| Model | *TP* | *TN* | *FP* | *FN* | *TPR* | *TNR* | *FPR* |
|---|---|---|---|---|---|---|---|
| LPI1 | 6345 | 25685 | 9442 | 1478 | 0.811 | 0.731 | 0.269 |
| LPI2 | 147 | 35063 | 64 | 7676 | 0.019 | 0.998 | 0.002 |
| LPI3 | 3737 | 31982 | 3145 | 4086 | 0.478 | 0.910 | 0.090 |
| LPIref | 5964 | 20826 | 14301 | 1859 | 0.762 | 0.593 | 0.407 |
| HAZ1 | 0 | 35127 | 0 | 7823 | 0.000 | 1.000 | 0.000 |
| HAZ2 | 0 | 35127 | 0 | 7823 | 0.000 | 1.000 | 0.000 |
| HAZ3 | 0 | 35127 | 0 | 7823 | 0.000 | 1.000 | 0.000 |
| ZHU1 | 1880 | 33483 | 1644 | 5943 | 0.240 | 0.953 | 0.047 |
| ZHU2 | 3135 | 31931 | 3196 | 4688 | 0.401 | 0.909 | 0.091 |
| ZHU3 | 2754 | 31017 | 4110 | 5069 | 0.352 | 0.883 | 0.117 |

**Table 6 – Summary of contingency table data and diagnostic scores for LPI models subject to sensitivity test without Juang et al.**
5 **(2005) correction factors being applied to *FS***

| Model | *TP* | *TN* | *FP* | *FN* | *TPR* | *TNR* | *FPR* |
|---|---|---|---|---|---|---|---|
| LPI2b | 610 | 34902 | 225 | 7213 | 0.078 | 0.994 | 0.006 |
| LPI3b | 5002 | 27136 | 7991 | 2821 | 0.639 | 0.773 | 0.227 |



**Table 7 – Model quality diagnostics and optimum threshold values for each model from ROC curves**

| Model | *AUC* | *J*-statistic | Threshold | *TPR* | *TNR* |
|---|---|---|---|---|---|
| LPI1 | 0.845 | 0.573 | 7 | 0.774 | 0.799 |
| LPI2 | 0.630 | 0.122 | 1 | 0.131 | 0.991 |
| LPI2b | 0.630 | 0.206 | 1 | 0.224 | 0.982 |
| LPI3 | 0.762 | 0.414 | 1 | 0.671 | 0.742 |
| LPI3b | 0.761 | 0.415 | 4 | 0.646 | 0.769 |
| LPIref | 0.748 | 0.366 | 6 | 0.689 | 0.678 |
| HAZ1 | 0.679 | 0.238 | 0.1 | 0.073 | 0.999 |
| HAZ2 | 0.608 | 0.316 | 0.1 | 0.134 | 0.997 |
| HAZ3 | 0.661 | 0.315 | 0.1 | 0.133 | 0.998 |
| ZHU1 | 0.753 | 0.355 | 0.1 | 0.556 | 0.799 |
| ZHU2 | 0.760 | 0.371 | 0.1 | 0.767 | 0.604 |
| ZHU3 | 0.718 | 0.306 | 0.1 | 0.712 | 0.594 |

**Table 8 – Coefficients of link function and summary of contingency table analysis for the two best performing LPI models**

| Model | $\beta_1$ | $B_0$ | *TPR* | *TNR* | *J*-statistic | *AUC* |
|---|---|---|---|---|---|---|
| LPI1 | 0.067 | -1.555 | 0.683 | 0.869 | 0.551 | 0.843 |
| LPI3b | 0.082 | -1.376 | 0.704 | 0.860 | 0.564 | 0.760 |

**Table 9 – Expected settlement amplitudes for liquefaction susceptibility zones from HAZUS methodology (NIBS, 2003)**

| Liquefaction susceptibility zone | Expected settlement amplitude (inches) |
|---|---|
| Very high | 12 |
| High | 6 |
| Moderate | 2 |
| Low | 1 |
| Very low | 0 |
| None | 0 |





**Table 10 – Summary statistics of PGDf$_V$ predictions for Darfield and Christchurch earthquakes from HAZUS models**

| Score | Observed | HAZ1 | HAZ2 | HAZ3 |
|---|---|---|---|---|
| Pearson R$^2$ | n/a | 0.064 | 0.051 | 0.058 |
| Mean | 0.118 | 0.003 | 0.008 | 0.005 |
| Minimum | 0.000 | 0.000 | 0.000 | 0.000 |
| Lower quartile | 0.051 | 0.000 | 0.000 | 0.000 |
| Median | 0.100 | 0.001 | 0.000 | 0.000 |
| Upper quartile | 0.162 | 0.004 | 0.004 | 0.004 |
| Maximum | 1.464 | 0.022 | 0.066 | 0.043 |
| Residual mean | n/a | -0.114 | -0.110 | -0.112 |
| RMSE | n/a | 0.146 | 0.142 | 0.144 |

**Table 11 – Summary statistics of PGDf$_H$ predictions for Darfield and Christchurch earthquakes from EPOLLS and HAZUS**
5  **models**

| Score | Observed | EPOLLS | HAZ1 | HAZ2 | HAZ3 |
|---|---|---|---|---|---|
| Pearson R$^2$ | n/a | 0.000 | 0.022 | 0.032 | 0.027 |
| Mean | 0.269 | 0.682 | 0.141 | 0.172 | 0.150 |
| Minimum | 0.001 | 0.149 | 0.000 | 0.000 | 0.000 |
| Lower quartile | 0.124 | 0.418 | 0.000 | 0.000 | 0.000 |
| Median | 0.206 | 0.748 | 0.084 | 0.050 | 0.067 |
| Upper quartile | 0.312 | 0.964 | 0.184 | 0.191 | 0.182 |
| Maximum | 3.856 | 1.989 | 1.872 | 3.205 | 2.443 |
| Residual mean | n/a | 0.413 | -0.128 | -0.096 | -0.118 |
| RMSE | n/a | 0.582 | 0.345 | 0.438 | 0.376 |