# Peer review of "Evaluating Simplified Methods for Liquefaction Assessment for Loss Estimation"

_Natural Hazards and Earth System Sciences, 2016_

## Referee Comment (RC1) · J. Douglas (Referee) · 21 Sep 2016

The authors test various methods for the assessment of liquefaction using data collected following two recent earthquakes in New Zealand. The study is within the scope of Natural Hazards and Earth System Sciences, it is generally well written, testing these methods using a large database of observations is a valuable exercise and the analysis appears to be carefully performed. Therefore, I recommend that this paper is accepted for publication but only after the following editorial changes are made.

1. Abstract, first sentence: This sentence is grammatically incorrect. In addition, it is probably too long to be easy understandable.

2. Abstract and throughout: "methods" or "procedures" are what is being talked about here. Therefore, these words should be used rather than "methodologies", which are

the principles that guide research practices.

3. Abstract and throughout: The word "data" is plural and hence the sentences should read "although these data may not" and "the input data are publicly".

4. P. 2, l. 8: This should probably read "both future risk assessments and post-event rapid response analyses".

5. P. 2, l. 9: Should it not be "liquefaction effects and physical damage" rather than "liquefaction risk and physical damage"?

6. P. 5, l. 12: USGS are not the only group to publish assessments of the ground motion following earthquakes so this organisation should only be given as an example.

7. P. 5, l. 13 (and elsewhere): Because of the high epistemic uncertainties in ground-motion prediction it is generally considered best practice to use a logic tree comprised of a set of ground-motion models rather than a single ground motion prediction equation. Hence I suggest slightly modifying this sentence.

8. P. 5, l. 15: I do not understand the comment "Although the use of Vs negates the requirement for ground investigation" because to assess Vs requires measurements on site, although they could be non-invasive (e.g. based on ambient noise approaches) as well as invasive (from boreholes). Vs30 could be estimated from geology or topographical slope, for example, but these would be uncertain and ideally should not be considered for site-specific analyses (e.g. Lemoine et al., Bulletin of the Seismological Society of America, 102, 2585-2599 2012).

9. P. 5, l. 17: I would change "extrapolate" to "estimate" or "interpolate" as extrapolation should be avoided.

10. P. 5, ll. 22-25: As noted in my comment 8 Vs30 from topographic slope (as provided by the USGS Global Vs30 Map Server) is uncertain because of the weak correlation between these variables. This should be commented on as a weakness of this approach.

11. P. 5, ll. 30: Boore et al. (Bulletin of the Seismological Society of America, 101, 3046-3059, 2011) update the relationships of Boore (2004) and other authors have proposed relationship for other parts of the world since 2004. I recommend making a comment that such relationships ideally should be regionally calibrated. Some checking that the equations of Boore (2004) are appropriate for New Zealand would be useful.

12. P. 5, Equations 8 and 9: It is not statistically correct to invert equations based on standard regression analysis (it would be acceptable if orthogonal regression had been used). I recommend adding a note that this inversion could be a source of uncertainty. Ideally a set of equations predicting Vs0-10 from Vs30 and Vs10-20 from Vs30 should have been derived based on regression in the correct direction.

13. P. 7, ll. 5-6: There seems to be a problem with the phrase "and for the other zones are given".

14. Section 2.3: Is it not circular to test this model on data from the Christchurch 2011 earthquake as data from this earthquake was used to develop it? I recommend adding a comment on this.

15. P. 8, l. 23: "compatring" should be "comparing" and "eartqhuake" should be "earthquake". Please spell check before manuscript submission.

16. P. 9, l. 3: What is the source of the moment magnitude of 6.2 for the 2011 earthquake? Both the USGS and Global CMT give Mw 6.1 for this event. Perhaps it is GeoNet. This should be stated.

17. Figure 2: What is the source of these contour maps?

18. P. 9, ll. 24-25: Are the results of SPT after the ground has liquefied appropriate to assess whether the ground is liquefiable? I would have thought that SPT values would be changed by liquefaction.

19. P. 10, ll. 1-5: Are Vs profiles at only 13 points sufficient to estimate Vs profiles

for the entire region? This may be appropriate if there are no changes in geology or topography across the city but it sounds too few for accurate results. A brief discussion on the uncertainties with this approach would be useful here. It would be useful to check the robustness of the interpolated profiles by removing one or more of the 13 profiles and comparing the results.

20. Section 4: It could help readability to split this section up into subsections for each of the tests.

21. Section 4: Why are the Zhu et al. (2015) models performing poorly when the Christchurch data was used in their development? There is a little discussion of this on pp. 14-15 but more discussion could be useful.

22. P. 5, l. 28: There is a problem with the grammar in the phrase "in both models though that the observed rates that are".

23. P. 17, l. 19, "tectonic uplift": Could it not also be "tectonic subsidence"? What about just saying "tectonic movements"?

24. P. 18, ll. 26-27: There is something missing from the sentence "To calculate duration, there are 19 strong-motion accelerograph stations in Christchurch that record ground motions at 0.02s intervals" as the stations are not there just to calculate duration.

25. P. 19, l. 18: It could be useful to say that even though the methods based on LPI are the best approaches tested that they still do not predict very well.

26. Figure 1: More information could be added to this map, e.g. the faults that ruptured in these earthquakes, the locations of the strong-motion stations used to estimate the durations, the locations of the 13 Vs profiles and the main areas of liquefaction (Figure 3). Currently, this map is not that useful and could be removed or combined with Figure 2 and/or Figure 3.

27. Table 4: It would be useful to combine this with Table 5.

28. Table 6: This could be added as an additional two lines to Table 5.

29. Table 7: Is there not space to include these results in Table 5?

30. Table 9: Could these numbers be added to Table 3 after conversion to SI units (e.g. cm)?

31. Tables 10 and 11: Give the units of the values reported here. Metres?

John Douglas 21st September 2016
* * *

---

## Referee Comment (RC2) · J. Douglas (Referee) · 22 Sep 2016

In addition to my comments yesterday, it would also significantly improve this manuscript to include some maps showing the areas predicted to liquefy by each of the methods. That would help understanding of the differences between the results rather than relying on statistical tests.

---

## Referee Comment (RC3) · Anonymous Referee #2 · 30 Oct 2016

This is a review of "Evaluating Simplified Methods for Liquefaction Assessment for Loss Estimation"

This paper is overall well written, interesting, and needing only minor revisions.

My comments, in no order of importance are as follows:

(a) Abstract. Please put more quantitative description of data/results into the abstract, not just qualitative.

(b) Section 1. Although there is a good 'motivation' and letting the reader know how the paper is organized in the last paragraph, can you refer to what the paper is about early on? Perhaps in or at the end of the paragraph a sentence that says "Here we investigate ....".

(c) Prediction. Please evaluate throughout the use of the word prediction (time, place, magnitude) if that is meant, or probabilistic forecasting. If prediction really is meant, then make this clear why, and to what degree.

(d) Testing. I am not a fan of the use of the word 'testing' in the natural hazard community. See the following paper for why: http://www.nssl.noaa.gov/users/brooks/public_html/feda/papers/Oreskes1.pdf

(e) Where possible, avoid acronyms in Figure captions/Table headers (or spell them out the first time) to make the paper a tad less 'jargon' rich. Figures and tables should be as stand-alone as possible so if someone uses them (without the paper) one can tell from the figure caption/table header what it is about. Particularly important for this is your figure comparing all the methods–you need to then state what all the acronyms mean.

(f) Would it be possible to provide an overview table of all the acronyms, and what data is being put into each one? This would be a nice 'tutorial' table that is more likely to be cited by people.

(g) Sensitivity of models to data input. It would be very nice to see more on how much the outputs (what you call prediction) are sensitive to slight changes in the inputs. Again, a comparison between different types of liquefaction models would be very useful.

Overall, enjoyed the paper, but think about how to make it accessible and more comparative/tutorial in nature where possible.

---

## Editor Comment (EC1) · B. D. Malamud (Editor) · 13 Nov 2016

13 November 2016

Dear Indranil Kongar, Tiziana Rossetto, and Sonia Giovinazzi:

We have now received two sets of reviewer comments on your manuscript "Evaluating Simplified Methods for Liquefaction Assessment for Loss Estimation". The three comments are largely positive, although do make suggestions.

I would like to invite you to respond to the reviewer comments, with for each item brought up stating how you would change the manuscript in response (or if not, a reason why not).

There is still time for other public comments to be made (until 6 December) so the

possibility of other items that you might need to reply to, after which point I will ask you to prepare a revised manuscript.

Regards, Bruce D Malamud [NHESS Executive Editor]

---

## Author Comment (AC1) · 4 Jan 2017

Reply to comments from referee 1 (by comment number), including the additional comment posted later

Reviewer #1: The authors test various methods for the assessment of liquefaction using data collected following two recent earthquakes in New Zealand. The study is within the scope of Natural Hazards and Earth System Sciences, it is generally well written, testing these methods using a large database of observations is a valuable exercise and the analysis appears to be carefully performed. Therefore, I recommend that this paper is accepted for publication but only after the following editorial changes are made

We thank the reviewer for his confirmation of the value of the research and recommendation for publication. Our responses to specific comments are included below.

1. Abstract, first sentence: This sentence is grammatically incorrect. In addition, it is probably too long to be easy understandable.

We shall amend to correct the grammar and to make it easier to understand

2. Abstract and throughout: "methods" or "procedures" are what is being talked about here. Therefore, these words should be used rather than "methodologies", which are the principles that guide research practices.

We shall replace "methodologies" with the word "method"

3. Abstract and throughout: The word "data" is plural and hence the sentences should read "although these data may not" and "the input data are publicly".

We shall amend the text to ensure that the word "data" is used as a plural

4. P. 2, l. 8: This should probably read "both future risk assessments and post-event rapid response analyses".

We shall amend accordingly

5. P. 2, l. 9: Should it not be "liquefaction effects and physical damage" rather than "liquefaction risk and physical damage"?

We shall amend accordingly

6. P. 5, l. 12: USGS are not the only group to publish assessments of the ground motion following earthquakes so this organisation should only be given as an example.

We shall add a note to make clear that the USGS is only one possible source of post-earthquake ground motion data

7. P. 5, l. 13 (and elsewhere): Because of the high epistemic uncertainties in ground-motion prediction it is generally considered best practice to use a logic tree comprised of a set of ground-motion models rather than a single ground motion prediction equation. Hence I suggest slightly modifying this sentence.

We shall modify this sentence to reflect the reviewer's comment

8. P. 5, l. 15: I do not understand the comment "Although the use of Vs negates the requirement for ground investigation" because to assess Vs requires measurements on site, although they could be non-invasive (e.g. based on ambient noise approaches) as well as invasive (from boreholes). Vs30 could be estimated from geology or topographical slope, for example, but these would be uncertain and ideally should not be considered for site-specific analyses (e.g. Lemoine et al., Bulletin of the Seismological Society of America, 102, 2585-2599 2012).

We thank the reviewer for highlighting this mistake. The original comment "Although the use of Vs negates the requirement for ground investigation" applies only to the Christchurch case study for which Vs data already exist. However, more generally the reviewer is correct to point out that Vs assessment does require ground investigation. In this instance, the intention is to talk about general approaches rather than a specific case study, and so we shall amend the text accordingly.

9. P. 5, l. 17: I would change "extrapolate" to "estimate" or "interpolate" as extrapolation should be avoided.

We shall amend the text as per the reviewer's suggestion

10. P. 5, ll. 22-25: As noted in my comment 8 Vs30 from topographic slope (as provided by the USGS Global Vs30 Map Server) is uncertain because of the weak correlation between these variables. This should be commented on as a weakness of this approach.

We shall add a comment to highlight this weakness

11. P. 5, ll. 30: Boore et al. (Bulletin of the Seismological Society of America, 101, 3046-3059, 2011) update the relationships of Boore (2004) and other authors have proposed relationship for other parts of the world since 2004. I recommend making a

comment that such relationships ideally should be regionally calibrated. Some checking that the equations of Boore (2004) are appropriate for New Zealand would be useful.

We acknowledge that ideally relationships should be regionally calibrated. However, the rationale of the paper is to test simplified methods to be used for loss estimation, principally by insurers. In this case, part of the definition of a 'simplified' method is that the relationships that underpin it already exist in the literature and no new model development is required. It should be noted that the relationships proposed by Boore et al. (2011) are not updates of the relationships of Boore (2004) – rather they are alternative relationships specific to Japan, while the original Boore (2004) relationships are specific to California. Nevertheless, we are happy to investigate the appropriateness of the Boore (2004) relationship to Christchurch, albeit on a small sample dataset of Vs profiles (13 sites), and will add this work to the revised manuscript.

12. P. 5, Equations 8 and 9: It is not statistically correct to invert equations based on standard regression analysis (it would be acceptable if orthogonal regression had been used). I recommend adding a note that this inversion could be a source of uncertainty. Ideally a set of equations predicting Vs0-10 from Vs30 and Vs10-20 from Vs30 should have been derived based on regression in the correct direction.

We shall add a note reflecting the reviewer's comment

13. P. 7, ll. 5-6: There seems to be a problem with the phrase "and for the other zones are given".

We shall amend the text to make this sentence clearer

14. Section 2.3: Is it not circular to test this model on data from the Christchurch 2011 earthquake as data from this earthquake was used to develop it? I recommend adding a comment on this.

We acknowledge that there is some circularity in this test and comment will be added

to that effect. However, it is worth noting that the datasets used to develop the model and test the model have not come from the same source and so may not be identical. Furthermore the original model has been calibrated to optimise estimation of the areal extent of liquefaction whereas in the case study exercise, site specific predictions are tested

15. P. 8, l. 23: "compatring" should be "comparing" and "eartqhuake" should be "earthquake". Please spell check before manuscript submission.

We shall amend these errors and run a spell check

16. P. 9, l. 3: What is the source of the moment magnitude of 6.2 for the 2011 earthquake? Both the USGS and Global CMT give Mw 6.1 for this event. Perhaps it is GeoNet. This should be stated.

The source is GeoNet and this will be stated

17. Figure 2: What is the source of these contour maps?

The source is the New Zealand Geotechnical Database and this will be stated in the caption

18. P. 9, ll. 24-25: Are the results of SPT after the ground has liquefied appropriate to assess whether the ground is liquefiable? I would have thought that SPT values would be changed by liquefaction.

Firstly we should note that there is a mistake here since the LPI values have been obtained from CPT, not SPT, data. Historically it has been thought that after liquefaction occurs, soils densify and increase their resistance to future liquefaction. However, Lees et al. (Soil Dynamics and Earthquake Engineering, 79, 304-314, 2015) conducted an analysis comparing CPT-based strength profiles and subsequent liquefaction susceptibility at sites in Christchurch both before and after the February 2011 earthquake, and concluded that no significant strengthening occurred and that the liquefaction risk in Christchurch after the earthquake remained the same as it was beforehand. The

study by Orense et al. (Geotech Eng J SEAGS & AGSSEA, 43(2), 8-17, 2012) came to similar conclusions and therefore our view is that the post-earthquake CPT data is appropriate for assessing liquefaction susceptibility

19. P. 10, ll. 1-5: Are Vs profiles at only 13 points sufficient to estimate Vs profiles for the entire region? This may be appropriate if there are no changes in geology or topography across the city but it sounds too few for accurate results. A brief discussion on the uncertainties with this approach would be useful here. It would be useful to check the robustness of the interpolated profiles by removing one or more of the 13 profiles and comparing the results.

We acknowledge that modelling Vs profiles for an entire city based on 13 samples has major limitations and the resulting estimations therefore carry significant uncertainty. We shall add a discussion of this as suggested. However we are not sure there is much value in carrying out the robustness check as suggested by the reviewer. Due to the aforementioned limitations, the check will almost certainly show that the interpolated profiles are not robust. However, testing the limitations/weaknesses of input data is not the point of this study. The purpose of this study is to investigate how a non-academic loss estimation analyst can use realistic and publicly available data and existing models to estimate liquefaction occurrence. A sample of 13 Vs profiles is a realistic dataset and the analysis in this paper has shown that despite its limitations, the resulting LPI model corresponding to it performs better than most other simplified models. There is little that an analyst can do if an input dataset is weak and nothing better is available – they can either make use of what they have or not conduct the analysis at all. It is not clear what value there is to be gained simply by proving that the input dataset is weak.

20. Section 4: It could help readability to split this section up into subsections for each of the tests.

We agree that Section 4 would benefit from being split into sub-sections

21. Section 4: Why are the Zhu et al. (2015) models performing poorly when the

Christchurch data was used in their development? There is a little discussion of this on pp. 14-15 but more discussion could be useful.

There are two possible reasons for this. One is that the model development and test datasets have come from different sources and so may not be identical. The second is that the Zhu et al. model has been optimised for quantification of the areal extent of liquefaction, whereas in the case study, site specific predictions are being tested

22. P. 5, l. 28: There is a problem with the grammar in the phrase "in both models though that the observed rates that are".

We shall amend the text to make this clearer

23. P. 17, l. 19, "tectonic uplift": Could it not also be "tectonic subsidence"? What about just saying "tectonic movements"?

We shall amend the text accordingly

24. P. 18, ll. 26-27: There is something missing from the sentence "To calculate duration, there are 19 strong-motion accelerograph stations in Christchurch that record ground motions at 0.02s intervals" as the stations are not there just to calculate duration.

We shall amend this sentence to make it clearer that duration can be measured from stations but this is not their primary purpose.

25. P. 19, l. 18: It could be useful to say that even though the methods based on LPI are the best approaches tested that they still do not predict very well.

We acknowledge this is a fair conclusion and will add a comment to this effect

26. Figure 1: More information could be added to this map, e.g. the faults that ruptured in these earthquakes, the locations of the strong-motion stations used to estimate the durations, the locations of the 13 Vs profiles and the main areas of liquefaction (Figure 3). Currently, this map is not that useful and could be removed or combined with Figure

2 and/or Figure 3.

We shall add the suggested information to Figure 1

27. Table 4: It would be useful to combine this with Table 5.

Table 4 relates to section 3 (Model test application) and Table 5 relates to section 4 (Results). Furthermore, Table 4 is referenced in the text much earlier than Table 5. We do not believe it would be appropriate to combine these tables

28. Table 6: This could be added as an additional two lines to Table 5.

We agree that these two tables can be combined

29. Table 7: Is there not space to include these results in Table 5?

It is important to stress the difference between Tables 5 and 7. The results in Table 5 are based on specified fixed thresholds, whereas the results in Table 7 are based on optimised thresholds. Therefore combining the two tables would necessitate the addition of two columns to identify the relevant threshold in each case. Therefore it may be possible to combine the tables if it can be published in landscape format - in portrait the combined table would appear very cluttered.

30. Table 9: Could these numbers be added to Table 3 after conversion to SI units (e.g. cm)?

We shall add this to Table 3

31. Tables 10 and 11: Give the units of the values reported here. Metres?

Yes, this is metres and the tables will be updated accordingly

32. In addition to my comments yesterday, it would also significantly improve this manuscript to include some maps showing the areas predicted to liquefy by each of the methods. That would help understanding of the differences between the results rather than relying on statistical tests.

We are happy to add the type of maps suggested by the reviewer, however due to the number of models tested in this paper, we believe it is appropriate to only include a selection of maps for the best performing models (e.g. the two best models or similar) in the revised manuscript

———————————————————

---

## Author Comment (AC2) · 4 Jan 2017

Reply to comments from referee 2 (by comment letter)

This is a review of "Evaluating Simplified Methods for Liquefaction Assessment for Loss Estimation". This paper is overall well written, interesting, and needing only minor revisions.

We thank the reviewer for his/her confirmation of the quality of the paper and recommendation for publication. Our responses to specific comments are included below.

(a) Abstract. Please put more quantitative description of data/results into the abstract, not just qualitative.

As per the reviewer's suggestion, we shall add more quantitative information to the

abstract relating to data and comparison of model performance

(b) Section 1. Although there is a good 'motivation' and letting the reader know how the paper is organized in the last paragraph, can you refer to what the paper is about early on? Perhaps in or at the end of the paragraph a sentence that says "Here we investigate ....".

We shall add a description of the paper's topic at the end of the first paragraph

(c) Prediction. Please evaluate throughout the use of the word prediction (time, place, magnitude) if that is meant, or probabilistic forecasting. If prediction really is meant, then make this clear why, and to what degree.

We acknowledge that the use of the word 'prediction' may be misleading here. Although the Hazus and Zhu models estimate liquefaction probabilities, the LPI models do not and furthermore, the final outputs being compared are deterministic estimates of liquefaction occurrence. Therefore 'probabilistic forecast' is not necessarily appropriate here. The model outputs are deterministic estimates of liquefaction occurrence conditional on an earthquake of specified place and magnitude having occurred and so a term of that nature, e.g. conditional estimate, may be more appropriate.

(d) Testing. I am not a fan of the use of the word 'testing' in the natural hazard community. See the following paper for why: http://www.nssl.noaa.gov/users/brooks/public_html/feda/papers/Oreskes1.pdf

We thank the reviewer for pointing us in the direction of this interesting and insightful article. We are happy to remove the word 'testing' and replace with an alternative term such as 'evaluation' or 'comparison' or similar

(e) Where possible, avoid acronyms in Figure captions/Table headers (or spell them out the first time) to make the paper a tad less 'jargon' rich. Figures and tables should be as stand-alone as possible so if someone uses them (without the paper) one can tell from the figure caption/table header what it is about. Particularly important for this

is your figure comparing all the methods–you need to then state what all the acronyms mean.

We acknowledge that the paper does contain a large number of acronyms and are happy to add more information to table and figure captions

(f) Would it be possible to provide an overview table of all the acronyms, and what data is being put into each one? This would be a nice 'tutorial' table that is more likely to be cited by people.

We are not clear what the reviewer means by this since a list of acronyms and list of variables are already provided in Tables 1 and 2 respectivley. If the reviewer feels that there are specific acronyms that require more detailed explanation, then it would be of assistance if the reviewer could specify which ones

(g) Sensitivity of models to data input. It would be very nice to see more on how much the outputs (what you call prediction) are sensitive to slight changes in the inputs. Again, a comparison between different types of liquefaction models would be very useful.

We are happy to carry out some sensitivity testing as suggested by the reviewer. A particularly useful sensitivity test may involve variation of shear wave velocity, since this is an input that is accompanied by significant uncertainty. However due to the number of models tested in this paper, we believe it is appropriate to only carry out the sensitivity testing for a selection of the best performing models (e.g. the two best models or similar) and we will add this to the revised manuscript

---

## Author Response (AR1)

**Ref. No.: NHESS-2016-281**
**Title: EVALUATING SIMPLIFIED METHODS FOR LIQUEFACTION ASSESSMENT FOR LOSS ESTIMATION**

5                                    **Author's Response to the Reviewers' Comments**

The authors would like to thank the reviewers for their efforts in reviewing this paper, the rapid review time and the constructive comments made. We have made a number of revisions to strengthen the paper in light of the reviewers' comments and considering the overall positive nature of the reviewers' comments to the first version

10 of the manuscript, we are confident that this improved version will be adequate for publication.

In addition, whilst working through parts of the analysis again, we noticed an error in the process used to derive Eq. 9 from the original equations of Boore (2004). Eq. 9 has been corrected in the revised text and this affects the results corresponding to models LPI3 and LPI3b as follows:

- With the new form of Eq. 9, model LPI3 now does meet the diagnostic score criteria of TPR > 0.5 and TNR > 0.5 when the threshold is 5, albeit with some bias towards negative forecasts.
- Model LPI3b still meets the criteria but with bias towards positive forecasts.
- The optimum threshold for model LPI3 is now 4, while the optimium threshold for model LPI3b is now 10
- Based on Youden's J-statistic, model LPI3 now outperforms model LPI3b
20
- In section 5, the liquefaction probability analysis of model LPI3b has been replaced by the equivalent analysis for model LPI3

Best Regards
Indranil Kongar, Tiziana Rossetto, Sonia Giovinazzi

25 **Reviewer #1:** The authors test various methods for the assessment of liquefaction using data collected following two recent earthquakes in New Zealand. The study is within the scope of Natural Hazards and Earth System Sciences, it is generally well written, testing these methods using a large database of observations is a valuable exercise and the analysis appears to be carefully performed. Therefore, I recommend that this paper is accepted for publication but only after the following editorial changes are made

*We thank the reviewer for his confirmation of the value of the research and recommendation for publication. Our responses to specific comments are included below.*

1. Abstract, first sentence: This sentence is grammatically incorrect. In addition, it is probably too long to be easy

35 understandable.

*This sentence has been amended to correct the grammar and to make ot easier to understand*

2. Abstract and throughout: "methods" or "procedures" are what is being talked about here. Therefore, these words should be used rather than "methodologies", which are the principles that guide research practices.

*We have replaced "methodologies" with the word "method"*

3. Abstract and throughout: The word "data" is plural and hence the sentences should read "although these data may not" and "the input data are publicly".

*We have amended the text to ensure that the word "data" is used as a plural*

4. P. 2, l. 8: This should probably read "both future risk assessments and post-event rapid response analyses".

15 *We have amended the text as suggested by the reviewer*

5. P. 2, l. 9: Should it not be "liquefaction effects and physical damage" rather than "liquefaction risk and physical damage"?

20 *We have amended the text as suggested by the reviewer*

6. P. 5, l. 12: USGS are not the only group to publish assessments of the ground motion following earthquakes so this organisation should only be given as an example.

25 *We have added a note to make clear that the USGS is only one possible source of post-earthquake ground motion data*

7. P. 5, l. 13 (and elsewhere): Because of the high epistemic uncertainties in ground-motion prediction it is generally considered best practice to use a logic tree comprised of a set of ground-motion models rather than a
30 single ground motion prediction equation. Hence I suggest slightly modifying this sentence.

*We have modified this sentence to reflect the reviewer's comment*

8. P. 5, l. 15: I do not understand the comment "Although the use of Vs negates the requirement for ground investigation" because to assess Vs requires measurements on site, although they could be non-invasive (e.g. based on ambient noise approaches) as well as invasive (from boreholes). Vs30 could be estimated from geology or topographical slope, for example, but these would be uncertain and ideally should not be considered for site-specific analyses (e.g. Lemoine et al., Bulletin of the Seismological Society of America, 102, 2585-2599 2012).

*We thank the reviewer for highlighting this mistake. The original comment "Although the use of Vs negates the requirement for ground investigation" applies only to the Christchurch case study for which Vs data already exist. However, more generally the reviewer is correct to point out that Vs assessment does require ground investigation. In this instance, the authors' intention was to talk about general approaches rather than a specific case study, and so we have amended the text accordingly.*

9. P. 5, l. 17: I would change "extrapolate" to "estimate" or "interpolate" as extrapolation should be avoided.

*We have amended the text as per the reviewer's suggestion*

10. P. 5, ll. 22-25: As noted in my comment 8 Vs30 from topographic slope (as provided by the USGS Global Vs30 Map Server) is uncertain because of the weak correlation between these variables. This should be commented on as a weakness of this approach.

*We have added a comment to highlight this weakness*

11. P. 5, ll. 30: Boore et al. (Bulletin of the Seismological Society of America, 101, 3046-3059, 2011) update the relationships of Boore (2004) and other authors have proposed relationship for other parts of the world since 2004. I recommend making a comment that such relationships ideally should be regionally calibrated. Some checking that the equations of Boore (2004) are appropriate for New Zealand would be useful.

*We acknowledge that ideally relationships should be regionally calibrated. However, the rationale of the paper is to test simplified methods to be used for loss estimation, principally by insurers. In this case, part of the definition*

*of a 'simplified' method is that the relationships that underpin it already exist in the literature and no new model development is required. It should be noted that the relationships proposed by Boore et al. (2011) are not updates of the relationships of Boore (2004) – rather they are alternative relationships specific to Japan, while the original Boore (2004) relationships are specific to California. Nevertheless, we have added an investigation of*
5   *the appropriateness of the Boore (2004) relationship to Christchurch in section 3.2.*

12. P. 5, Equations 8 and 9: It is not statistically correct to invert equations based on standard regression analysis (it would be acceptable if orthogonal regression had been used). I recommend adding a note that this inversion could be a source of uncertainty. Ideally a set of equations predicting Vs0-10 from Vs30 and Vs10-20 from Vs30
10   should have been derived based on regression in the correct direction.

*The authors are not aware of any relationships that are derived in the correct direction, and as mentioned in the previous comment, the development of new relationships contradicts the objectives of this study. We have added a note to section 2.1 reflecting the reviewer's comment regarding uncertainty due to inversion.*

13. P. 7, ll. 5-6: There seems to be a problem with the phrase "and for the other zones are given".

*We have amended the text to make this sentence clearer*

20   14. Section 2.3: Is it not circular to test this model on data from the Christchurch 2011 earthquake as data from this earthquake was used to develop it? I recommend adding a comment on this.

*We acknowledge that there is some circularity in this test. However, it is worth noting that the datasets used to develop the model and test the model have not come from the same source and so may not be identical.*
25   *Furthermore the original model has been calibrated to optimise estimation of the areal extent of liquefaction whereas in the case study exercise, site specific predictions are tested. A comment to this effect has been added at the end of section 2.3.*

15. P. 8, l. 23: "compatring" should be "comparing" and "eartqhuake" should be "earthquake". Please spell check
30   before manuscript submission.

*We have run a spell check and corrected these errors*

16. P. 9, l. 3: What is the source of the moment magnitude of 6.2 for the 2011 earthquake? Both the USGS and Global CMT give Mw 6.1 for this event. Perhaps it is GeoNet. This should be stated.

*The source is indeed GeoNet and the reference (GNS Science, 2014) has been added at the end of the relevant sentence.*

17. Figure 2: What is the source of these contour maps?

*The source is the Canterbury Geotechnical Database and the reference has been added in the caption*

18. P. 9, ll. 24-25: Are the results of SPT after the ground has liquefied appropriate to assess whether the ground is liquefiable? I would have thought that SPT values would be changed by liquefaction.

*Firstly we should note that there is a mistake here since the LPI values have been obtained from CPT, not SPT, data and so this has been corrected. Historically it has been thought that after liquefcation occurs, soils densify and increase their resistance to future liqueaction. However, Lees et al. (Soil Dynamics and Earthquake Engineering, 79, 304-314, 2015) conducted an analysis comparing CPT-based strength profiles and subsequent liquefaction susceptibility at sites in Christchurch both before and after the February 2011 earthquake, and concluded that no significant strengthening occurred and that the liquefaction risk in Christchurch after the earthquake remained the same as it was beforehand. The study by Orense et al. (Geotech Eng J SEAGS & AGSSEA, 43(2), 8-17, 2012) came to similar conclusions and therefore our view is that the post-earthquake CPT data is appropriate for assessing liquefaction susceptibility. This explanantion has been added to the text.*

19. P. 10, ll. 1-5: Are Vs profiles at only 13 points sufficient to estimate Vs profiles for the entire region? This may be appropriate if there are no changes in geology or topography across the city but it sounds too few for accurate results. A brief discussion on the uncertainties with this approach would be useful here. It would be useful tocheck the robustness of the interpolated profiles by removing one or more of the 13 profiles and comparing the results.

*We acknowledge that modelling Vs profiles for an entire city based on 13 samples has major limitations and the resulting estimations therefore carry significant uncertainty. We have added a comment to this effect. However we are not sure there is much value in carrying out the robustness check as suggested by the reviewer. Due to the aforementioned limitations, the check will almost certainly show that the interpolated profiles are not robust. However, testing the limitations/weaknesses of input data is not the point of this study. On the contrary, the point of this study is to show what can be done with weak/limited datasets and more specifically, to investigate how a non-academic loss estimation analyst can use realistic, publicly avaiable (but weak) data, together with existing models, to estimate liquefaction occurrence. The sample of 13 Vs profiles, acquired from an academic journal paper, is excatly the type of realistic dataset that a non-academic analyst can expect to have access to and the case study analysis has shown that despite its limitations, the resulting LPI model that corresponds to it performs better than most other simplified models. There is little that an analyst can do if an input dataset is weak and nothing better is available – they can either make use of what they have or not conduct the analysis at all. It is not clear what value there is to be gained simply by proving that the input dataset is weak.*

20. Section 4: It could help readability to split this section up into subsections for each of the tests.

*We agree that Section 4 would benefit from being split into sub-sections and this has been done*

21. Section 4: Why are the Zhu et al. (2015) models performing poorly when the Christchurch data was used in their development? There is a little discussion of this on pp. 14-15 but more discussion could be useful.

*There are two possible reasons for this. One is that the model development and test datasets have come from different sources and so may not be identical. The second, which has already been identified in the discussion, is that the Zhu et al. model has been optimised for quantification of the areal extent of liquefaction, whereas in the case study, site specific predictions are being tested. In light of this, the authors are keen to stress that the results of the case study do not contradict or invalidate the findings of Zhu et al. (2015) and a comment has been added to this effect.*

22. P. 5, l. 28: There is a problem with the grammar in the phrase "in both models though that the observed rates that are".

*The sentence has been amended to make this clearer*

23. P. 17, l. 19, "tectonic uplift": Could it not also be "tectonic subsidence"? What about just saying "tectonic movements"?

*We have amended the text accordingly*

24. P. 18, ll. 26-27: There is something missing from the sentence "To calculate duration, there are 19 strong-motion accelerograph stations in Christchurch that record ground motions at 0.02s intervals" as the stations are
10 not there just to calculate duration.

*We have amended this sentence to make it clearer that duration can be measured from stations but this is not their primary purpose.*

15 25. P. 19, l. 18: It could be useful to say that even though the methods based on LPI are the best approaches tested that they still do not predict very well.

*We acknowledge this is a fair conclusion and have added a comment to this effect*

20 26. Figure 1: More information could be added to this map, e.g. the faults that ruptured in these earthquakes, the locations of the strong-motion stations used to estimate the durations, the locations of the 13 Vs profiles and the main areas of liquefaction (Figure 3). Currently, this map is not that useful and could be removed or combined with Figure 2 and/or Figure 3.

25 *We have added the suggested information to Figure 1*

27. Table 4: It would be useful to combine this with Table 5.

*Table 4 relates to section 3 (Model test application) and Table 5 relates to section 4 (Results). Furthernore,*
30 *Table 4 is referenced in the text much earlier than Table 5. We do not believe it would be appropriate to combine these tables*

28. Table 6: This could be added as an additional two lines to Table 5.

*We agree and these two tables have been combined*

29. Table 7: Is there not space to include these results in Table 5?

*It is important to stress the difference between Tables 5 and 7. The results in Table 5 are based on specified fixed thresholds, whereas the results in Table 7 are based on optimised thresholds. Therefore combining the two tables*
10 *would necessitate the addition of two columns to identify the relevant treshold in each case. Therefore it may be possible to combine the tables if it can be published in landscape format - in portrait the combined table would appear very cluttered. At this stage we feel it is better and clearer to keeo these two tables distinct.*

30. Table 9: Could these numbers be added to Table 3 after conversion to SI units (e.g. cm)?

*We have added this data to Table 3*

31. Tables 10 and 11: Give the units of the values reported here. Metres?

20 *Yes, this is metres and the tables have been updated accordingly*

32. In addition to my comments yesterday, it would also significantly improve this manuscript to include some maps showing the areas predicted to liquefy by each of the methods. That would help understanding of the differences between the results rather than relying on statistical tests.

*We have added the type of maps suggested by the reviewer, however due to the number of models tested in this paper, we believed it was appropriate to only include a selection of maps for the best performing models (LPI1, LPI3, ZHU1 and ZHU2)*

**Reviewer #2**: This is a review of "Evaluating Simplified Methods for Liquefaction Assessment for Loss Estimation". This paper is overall well written, interesting, and needing only minor revisions.

*We thank the reviewer for his/her confirmation of the quality of the paper and recommendation for publication. Our responses to specific comments are included below.*

5  My comments, in no order of importance are as follows:

(a) Abstract. Please put more quantitative description of data/results into the abstract, not just qualitative.

*As per the reviewer's suggestion, we have added more quantitative information to the abstract relating to data*
10  *and comparison of model performance*

(b) Section 1. Although there is a good 'motivation' and letting the reader know how the paper is organized in the last paragraph, can you refer to what the paper is about early on? Perhaps in or at the end of the paragraph a sentence that says "Here we investigate ....".

*We have added a short description of the paper's topic at the end of the second paragraph of the introduction*

(c) Prediction. Please evaluate throughout the use of the word prediction (time, place, magnitude) if that is meant, or probabilistic forecasting. If prediction really is meant, then make this clear why, and to what degree.

*We do not use the word 'prediction' in the sense that the reviewer defines it and therefore acknowledge that the use of the word 'prediction' may be misleading and other terms should be used. Although the Hazus and Zhu models estimate liquefaction probabilities, the LPI models do not and furthermore, the final outputs from all the models are deterministic estimates of liquefaction occurrence, which are conditional on an earthquake of*
25  *specified place and magnitude having occurred. Therefore the description of the forecasts as probabilistic is not necessarily appropriate here and we have therefore amended the text to replace instances of the word 'prediction' with either 'forecast' or 'estimate'.*

(d) Testing. I am not a fan of the use of the word 'testing' in the natural hazard community. See the following
30  paper for why: http://www.nssl.noaa.gov/users/brooks/public_html/feda/papers/Oreskes1.pdf

*We thank the reviewer for pointing us towards this interesting and insightful article, which offers the view that words such as 'testing' are misused in the scientific community to give a false view of model perfection. With respect to our case study, we are clear throughout our paper that the objective is to compare the relative quality/performance of the range of models with respect to a set of observations. We are not claiming that any one model is 'right' and believe we are using the word 'testing' correctly. Nevertheless, it may be open to misinterpretation and for that reason, we have removed the words 'test' and 'testing' where appropriate (i.e. not where it refers to the name of a test) and have replaced with a suitable alternative. Most commonly we have used 'compare' or 'comparison' to emphasise this aspect of the study.*

(e) Where possible, avoid acronyms in Figure captions/Table headers (or spell them out the first time) to make the paper a tad less 'jargon' rich. Figures and tables should be as stand-alone as possible so if someone uses them (without the paper) one can tell from the figure caption/table header what it is about. Particularly important for this is your figure comparing all the methods–you need to then state what all the acronyms mean.

*We acknowledge that the paper does contain a large number of acronyms and have added more information to some figure captions and table headers. Where tables/figures contain model acronyms, writing out the full model names is no more informatve and writing out the full descriptions is impractical and so instead we have added a reference to Table 4 in the caption.*

(f) Would it be possible to provide an overview table of all the acronyms, and what data is being put into each one? This would be a nice 'tutorial' table that is more likely to be cited by people.

*We are not entirely clear what the reviewer means by this since a list of acronyms and list of variables are already provided in Tables 1 and 2 respectivley. However we have added a column to Table 2 identifying the variable input data that goes into evaluating each of the listed variables.*

(g) Sensitivity of models to data input. It would be very nice to see more on how much the outputs (what you call prediction) are sensitive to slight changes in the inputs. Again, a comparison between different types of liquefaction models would be very useful.

*We have carried out two sensitivity tests involve variation of shear wave velocity and peak ground acceleration. However due to the number of models tested in this paper, we have only carried out the sensitivity testing for two of the best performing models (e.g. the two best models or similar) and have added a summary to the revised manuscript. We would also like to point out that the optimisation of threshold, already carried out for the original paper, is also a type of sensitivity test.*

**Evaluating Simplified Methods for Liquefaction Assessment for Loss Estimation**

Indranil Kongar[1], Tiziana Rossetto[1], Sonia Giovinazzi[2]

[1]Earthquake and People Interaction Centre (EPICentre), Department of Civil, Environmental and Geomatic Engineering, University College London, London, WC1E 6BT, United Kingdom

[2]Department of Civil and Natural Resources Engineering, University of Canterbury, Christchurch, 8140, New Zealand

*Correspondence to*: Indranil Kongar (ucfbiko@ucl.ac.uk)

**Abstract.** Currently, some catastrophe models used by the insurance industry account for liquefaction  by applying a  factor to shaking-induced losses. The factor is based only on local liquefaction susceptibility and  the need for a more sophisticated approach to incorporating the effects of liquefaction in loss models . This study compares eleven unique models, each based on one of three principal simplified liquefaction assessment methods: liquefaction potential index (LPI) calculated from shear-wave velocity; the HAZUS software method; and a method created specifically to make use of USGS remote sensing data. Data from the September 2010 Darfield and February 2011 Christchurch earthquakes in New Zealand are used to compare observed liquefaction occurrences to  forecasts from these models using binary classification performance measures. The analysis shows that the best performing model is the LPI calculated using known shear-wave velocity profiles, which correctly forecasts 78% of sites where liquefaction occurred and 80% of sites where liquefaction did not occur, when the threshold is set at 7. However,  these data may not always be available to insurers. The next best model is also based on LPI, but uses shear-wave velocity profiles simulated from the combination of USGS $V_{S30}$ data and empirical functions that relate $V_{S30}$ to average shear-wave velocities at shallower depths. This model correctly forecasts 58% of sites where liquefaction occurred and 84% of sites where liquefaction did not occur, when the threshold is set at 4. These scores increase to 78% and 86% respectively when forecasts are based on liquefaction probabilities that are empirically related to the same values of LPI. This model is potentially more useful for insurance since the input data are publicly available. HAZUS models, which are commonly used in studies where no local model is available, perform poorly and incorrectly forecast 87% of sites where liquefaction occurred, even at optimal thresholds. This paper also considers two models (HAZUS and EPOLLS) for prediction of the scale of liquefaction in terms of permanent ground deformation but finds that both models perform poorly, with correlations between observations and forecasts lower than 0.4 in all cases. Therefore these models potentially provide negligible additional value to loss estimation analysis outside of the regions for which they have been developed.

**1 Introduction**

The recent earthquakes in Haiti (2010), Canterbury, New Zealand (2010-11) and Tohoku, Japan (2011) highlighted the significance of liquefaction as a secondary hazard of seismic events and the significant damage that it can cause to buildings and infrastructure. However, the insurance sector was caught out by these events, with catastrophe models underestimating the extent and severity of liquefaction that occurred (Drayton and Verdon, 2013). A contributing factor to this is that the method used by some catastrophe models to account for liquefaction is based only on liquefaction susceptibility, a qualitative parameter that considers only surficial geology characteristics. Furthermore, losses arising from liquefaction are estimated by adding an amplifier to losses estimated due to building damage caused by ground shaking (Drayton and Vernon 2013). There is a paucity of past event data on which to calibrate an amplifier and consequently, significant losses from liquefaction damage will only be estimated if significant losses are already estimated from ground shaking, whereas it is known that liquefaction can be triggered at relatively low ground shaking intensities (Quigley et al., 2013).

Therefore there is scope within the insurance and risk management sectors to adopt more sophisticated approaches for forecasting liquefaction for both future risk assessments and post-event rapid response analyses. It is also important to develop a better understanding of the correlation between liquefaction effects and physical damage of the built environment, similar to the fragility functions that are used to  estimate damage associated with ground shaking. This is particularly the case for critical infrastructure systems since, whilst liquefaction is less likely than ground shaking to be responsible for major building failures (Bird and Bommer, 2004), it can have a major impact on lifelines such as roads, pipelines and buried cables. Loss of power and reduction in transport connectivity are major factors affecting the resilience of business organizations in response to earthquakes as they can delay the recommencement of normal operations. Evaluating the seismic performance of infrastructure is therefore critical to understanding indirect economic losses caused by business interruption and to achieve this it is necessary to assess the liquefaction risk in addition to that posed by ground shaking. Therefore in this paper we investigate the performance of a range of models that can be applied to forecast the occurrence and scale of liquefaction based on simple and accessible input datasets. The performances are evaluated by comparing model forecasts to observations from the 2010-11 Canterbury earthquake sequence,

[revised manuscript text omitted]
 various online sources, with one example being the USGS ShakeMaps (USGS, 2014a) However if they are not, then  it would be necessary to apply engineering judgment in the selection of  appropriate ground motion prediction equations (either a single equation or multiple equations applied in a logic tree). The *LPI* model also requires water table depth and soil unit weights. If these are not known exactly, engineering judgment needs to be applied to estimate these based on information in existing literature. For the specific case study presented in this paper, some $V_S$ data are available from published sources.  However, more generally $V_S$ data  is not  in the public domain and would require ground investigation to acquire. Even in cases where $V_S$ data are available,  across the entire study area, thus requiring geostatistical techniques to  interpolate. Consequently this method may only be applicable in a small number of study areas.

[revised manuscript text omitted]

**3 Model test assessment application**

This section summarises the procedure for compatring the model predictions forecasts to observations from the Canterbury eartqhuakeearthquake sequence. A brief description is provided of the liquefaction observation dataset and the additional datasets accessed in order to provide the required inputs to the nine models. This is followed by a discussion on the conversion of quantitative model outputs to categorical liquefcationliquefaction predictions forecasts and an explanantionexplanation of the test diagnostics used to assess model performance.

**3.1 Liquefaction observations**

The methods described in the previous section are tested compared for two case studies from the Canterbury earthquake sequence: the $M_W$ 7.1 Darfield earthquake on September 4th 2010 and the $M_W$ 6.2 Christchurch earthquake on February 22nd 2011 (GNS Science, 2014), as identified in Figure 1Figure 1. The corresponding peak horizontal ground acceleration contours for each earthquake are shown in Figure 2.

Surface liquefaction observation data have been obtained from two sources: ground investigation data provided directly from Tonkin & Taylor, geotechnical consultants to the New Zealand Earthquake Commission (EQC) (van Ballegooy et al., 2014) and maps stored within the Canterbury Geotechnical Database (CGD, 2013a), an online repository of geotechnical data and reports for the region set up by EQC for knowledge sharing after the earthquakes. The data provided by Tonkin & Taylor include records from over 7,000 geotechnical investigation sites across Christchurch. After each earthquake, a land damage category is attributed to each site, representing a qualitative assessment of the scale of liquefaction observed. There are six land damage categories, but since this study only investigates liquefaction triggering, the categories are converted to a binary classification of liquefaction occurrence. These data are supplemented by the maps from the CGD which show the areal extent of the same land damage categories. To ensure equivalence in the study, all models are applied to the same study area for each earthquake, which is the region for which the input data for all models are available. The study area is divided into a grid of 100m x 100m squares, generating 25,100 observation sites. It is noted however that at some locations within Christchurch, no liquefaction observations are available so these sites are excluded from the subsequent analysis. As a result, the study area consists of 20,147 sites for the Darfield earthquake and 22,803 sites for the Christchurch earthquake. The observations from the two events are shown in Figure 3.

**3.2 Forecast model inputs**

This study includes three implementations of the LPI model: 1) using known $V_S$ profiles (referred to as LPI1 in this paper); 2) using $V_{S30}$ as a proxy for $V_S$ (LPI2); and 3) using 'realistic' $V_S$ profiles simulated from $V_{S30}$ and the Boore (2004) functions (LPI3). The geotechnical investigation data provided by Tonkin & Taylor also include values of $LPI$ calculated at each site from CPT data rather than $V_S$. Although this approach is not feasible for insurers, for reference its forecasting power is also compared here. This implementation is referred to as LPIref. Historically it has been thought that after liquefaction occurs, soils densify and increase their resistance to future liquefaction. However, Lees et al. (2015) conducted an analysis comparing CPT-based strength profiles and subsequent liquefaction susceptibility at sites in Christchurch both before and after the February 2011 earthquake. They concluded that no significant strengthening occurred and that the liquefaction risk in Christchurch after the earthquake remained the same as it was beforehand. The study by Orense et al. (2012) came to similar conclusions and therefore for the purposes of this case study, post-earthquake CPT data is appropriate for assessing liquefaction susceptibility.

A water table depth of 2m has been assumed across Christchurch, reflecting the averages described by Giovinazzi et al. (2011) – 0-2m in the eastern suburbs and 2-3m in the western suburbs – and soil unit weights of 17kPa above the water table and 19.5kPa below the water table are assumed, as suggested by Wotherspoon et al. (2014). $V_{S30}$ data for LPI2 and LPI3 are taken from the USGS web server, with point estimates on an approximately 674m grid.

Wood et al. (2011) have published $V_S$ profiles for 13 sites across Christchurch obtained using surface wave testing methods. These sites are identified in Figure 1. In GIS, the profiles are converted to point data for each 1m depth increment from 0-20m, so that each point represents the $V_S$ at that site for a single soil layer and there are a total of 13 points for each soil

5     layer. Ordinary kriging (with log transformation to ensure non-negativity) is applied to the points in each soil layer to create interpolated $V_S$ raster surfaces for each layer. Interpolation over a large area from such a small number of points is likely to result in estimations carrying significant uncertainty. However from the perspective of commercial loss estimation, this is typical of the type of data that an analyst may be required to work with and so there is value in investigating its efficacy. Whilst Andrus and Stokoe (2000) advise that the maximum $V_{S1}$ can range from 200-215m/s depending on fines content,

10    subsequent work by Zhou and Chen (2007) indicates that the maximum $V_{S1}$ could range between 200-230m/s. In the absence of specific fines content data, a median value of 215m/s is assumed to be the maximum. In practice, a soil layer may have a value of $V_{S1}$ below this threshold but not be liquefiable because the soil is not predominantly clean sand. Because of the regional scale of this analysis though, site-specific soil profiles (as distinct from $V_S$ profile) are not taken into account in determining whether a soil layer is liquefiable. Goda et al. (2011) suggest the use of 'typical' soil profiles to determine the

15    liquefaction susceptibility of a soil layer at a regional scale. Borehole data at sites close to the 13 $V_S$ profile sites are available from the Canterbury Geotechnical Database (CGD, 2013c). These indicate that in the eastern suburbs of Christchurch, soil typically consists predominantly of clean sand to 20m depth, with some layers of silty sand. On the western side of Christchurch however there is an increasing mix of sand, silt and gravel in soil profiles, particularly at depths down to 10m. Therefore it is possible, particularly in western suburbs, that the calculated $V_{S1}$ values may indicate liquefiable soil layers

20    when they are in fact not, which would lead to overestimation of *LPI* and the extent of liquefaction.

For the implementation of model LPI3, it could be argued that rather than using the Boore (2004) relationships to estimate $V_S$ profiles at shallower depths from $V_{S30}$, the local $V_S$ data published by Wood et al. (2011) could be used to develop a locally calibrated model. This would be preferable from a purely scientific perspective. However, the purpose of this study is

25    to investigate the potential for a simple 'global' model for commercial application, and this is defined in part as a model that makes use of methods already in the literature and does not require additional model development. Nevertheless, when using existing models it is useful to assess their applicability to a study area, and the $V_S$ profiles published by Wood et al. (2011) can be used to assess the suitability of the Boore (2004) relationships in Christchurch. Figure 4 shows plots of $V_{S30}$ against $V_{S10}$ and $V_{S20}$ as calculated from the observed profiles and compares these to the Boore (2004) functions. The plots show that

30    that the relationships exhibit a small bias towards the underestimation of $V_{S30}$. When inverted, the application of these relationships to Christchurch may therefore result in the overestimation of $V_S$ at shallower depths and therefore underestimate liquefaction occurrence. However, the majority of observed values are within the 95% confidence intervals and so the relationships can be deemed to be applicable.

[revised manuscript text omitted]

The ZHU1 and ZHU2 models perform reasonably with *AUC* values and *J*-statistics slightly lower than the LPI3 models, but the optimum thresholds are at the minimum of the tested range that has been investigated, confirming the degree to which these models under-predict estimate liquefaction occurrence. The ZHU2 model also meets the *TPR* and *TNR* criteria with a threshold value of 0.2, albeit with a greater prediction forecast bias (*J*-statistic = 0.370, *TPR* = 0.555, *TNR* = 0.815). The ZHU3 model, despite being specific to Christchurch, does not perform as well as ZHU1 or ZHU2. There are potential

reasons for this anomaly, such as that  the ZHU models were calibrated to preserve the extent of liquefaction rather than to make site-specific forecasts or because the data used to develop the models has not come from the same source as the observation data used for comparison. Therefore  these results do not contradict or invalidate the original findings of Zhu et al. (2015).

The absolute quality of models is  evaluated by calculating *MCC*. In the preceding analysis, the best performing model is LPI1 and this has a value of *MCC* = 0.48. The correlation is only moderate, but nevertheless indicates that the model is better than random guessing. As part of a rapid assessment or desktop study for insurance purposes, this may be sufficient. LPI3 and LPI3b have *MCC* = 0.380 and 0.358 respectively, whilst LPIref has *MCC* = 0.29.

**4.3 Mapping of model forecasts**

The maps in Figure 6 and Figure 7 show how forecasts of liquefaction occurrence, relating to the Darfield and Christchurch earthquakes respectively, are distributed across the city for four of the best performing models identified in Table 6: LPI1, LPI3, ZHU1 and ZHU2. Figure 3 shows that a greater extent of liquefaction was observed in the Christchurch earthquake than in the Darfield earthquake and this is reflected by all four models represented in Figure 6 and Figure 7. However for both earthquakes, each of the models forecasts a greater extent of liquefaction than was observed. In the Darfield earthquake, most of the liquefaction was observed in the north and east of the city. Whilst to some degree, this spatial distribution is matched by model LPI1, the remaining models do not represent the observed distribution well, with In particular the models ZHU1 and ZHU2 estimate a greater proportion of liquefaction in the south of the city. In the Christchurch earthquake, liquefaction was mostly observed in the eastern suburbs of the city. All the models forecast the majority of liquefaction to occur in these areas, although model ZHU2 forecasts more liquefaction occurring in western suburbs than actually occurred, while model ZHU1 forecasts no liquefaction occurring to the west of the city at all. The spatial distributions of the forecasts from the LPI models exhibit only limited accuracy, yet they are better than the forecasts from the two ZHU models. This can be explained partly by the fact the LPI method is designed for site-specific estimation, whereas the ZHU models have been calibrated to optimise the extent rather than the location of liquefaction.

**4.4 Sensitivity test – $V_{S30}$**

The sensitivity of the forecasts to variation in $V_{S30}$ is assessed for models LPI3 and ZHU1. LPI3 is the best performing model that requires $V_{S30}$ and ZHU2 is the best performing ZHU model. The forecasting procedure and contingency table analysis for the two models are repeated for two scenarios, one where $V_{S30}$ is decreased by 10% at all sites and one where $V_{S30}$ is increased by 10% at all sites.

In the scenario where $V_{S30}$ is decreased, the *TPR* for model LPI3 increases to 0.819 with a threshold of 4 (the optimised threshold from Table 6), while the *TNR* decreases to 0.536, effectively reversing the bias demonstrated by the original

model. The *J*-statistic reduces significantly to 0.356 indicating lower performance than the original model. With the new $V_{S30}$ values, the optimised threshold increases to 9, with *J*-statistic = 0.426, which is higher than the original model, *TPR* = 0.654 and *TNR* = 0.773. When $V_{S30}$ is increased, *TPR* = 0.308 with a threshold of 4, which is lower than criterion for good performance (*TPR* > 0.5), *TNR* = 0.974. This demonstrates a strengthening of the negative bias in the original model and
5 poor performance since the *J*-statistic reduces to 0.282. The optimum threshold changes to 1, yet even with this threshold, while the *J*-statistic improves to 0.388, *TPR* = 0.489, which is still below the performance criterion. These results show that LPI3 forecasts are sensitive to variation in $V_{S30}$. Therefore, although currently the optimum LPI3 threshold for Christchurch has been identified as 4, if in the future more accurate $V_{S30}$ becomes available, then the analysis presented in this paper should be repeated to recalibrate model LPI3 with a new optimum threshold.

Model ZHU2 experiences much smaller changes as a result of changes to $V_{S30}$. When $V_{S30}$ is decreased, and with a threshold of 0.1 (the optimised threshold from Table 6), *TPR* = 0.820, *TNR* = 0.532 and *J*-statistic = 0.352. When $V_{S30}$ is increased, *TPR* = 0.700, *TNR* = 0.662 and *J*-statistic = 0.362. For both scenarios all performance criteria are met and there only small reductions in *J*-statistic. When the models are optimised, the thresholds change to 0.25 for the decrease (*J*-statistic = 0.370)
15 and to 0.15 for the increase (*J*-statistic = 0.368). These results suggest that ZHU2 forecasts are relatively stable in response to variations in $V_{S30}$, but if more accurate $V_{S30}$ becomes available in the future, then some performance improvement can be achieved through recalibration of the optimum threshold.

**4.5 Sensitivity test – *PGA**

The sensitivity of the forecasts to uncertainty in *PGA* measurements is also assessed for models LPI3 and ZHU1. In the two
20 sensitivity test scenarios, The forecasting procedure and contingency table analysis are repeated for two scenarios, where *PGA* is decreased by 10% at all sites and where *PGA* is increased by 10% at all sites.

In the scenario where *PGA* is decreased, the *TPR* for model LPI3 decreases to 0.503 with a threshold of 4, while the *TNR* increases to 0.905 and there is only a small reduction in *J*-statistic to 0.408. The optimised threshold decreases to 2, with *J*-
25 statistic = 0.424, which is higher than the original model, *TPR* = 0.594 and *TNR* = 0.830. When *PGA* is increased, *TPR* = 0.652 with a threshold of 4 and *TNR* = 0.765, with corresponding *J*-statistic = 0.417. The optimum threshold changes to 6, with *J*-statistic = 0.419, *TPR* = 0.576 and *TNR* = 0.843. In general, changes in *PGA* do affect the scores but in all cases, the changes are relatively small, particularly with respect to the *J*-statistic, and the performance criteria are still met.

30 Model ZHU2 also experiences small changes as a result of changes to *PGA*. When *PGA* is decreased, and with a threshold of 0.1, *TPR* = 0.725, *TNR* = 0.637 and *J*-statistic = 0.362. When *PGA* is increased, *TPR* = 0.798, *TNR* = 0.574 and *J*-statistic = 0.372, which is a small increase over the original model. For both scenarios all performance criteria are met and there only small changes to *J*-statistic. When the models are optimised, the threshold changes to 0.2 for the decrease scenario (*J*-

statistic = 0.369), but for the increase scenario, the optimum threshold is still 0.1. These results suggest that both LPI3 and ZHU2 forecasts are relatively stable in response to variations in *PGA* and so while small uncertainties in *PGA* measurements will change the rates of true positive and true negative forecasts, overall performance in terms of *J*-statistic remains similar.

**5 Probability of liquefaction**

[revised manuscript text omitted]

Zhu, J., Daley, D., Baise, L. G., Thompson, E. M., Wald, D. J. and Knudsen, K. L., (2015), A geospatial liquefaction model for rapid response and loss estimation, Earthq. Spectra, 31(3), 1813-1837.

[Figure]

[Figure]

**Figure 1 – Locations of epicentres and fault planes (Beaven et al., 2012) of the Darfield and Christchurch earthquakes, strong motion stations from which recordings are used to estimate shaking durations and locations at which shear wave velocity ($V_s$) profiles are known (Wood et al., 2011). Note that locations of $V_s$ profiles coincide with strong motion stations. in relation to the Christchurch urban area and central business district.**

[Figure]

**Figure 2 – Contours of peak horizontal ground acceleration (PGA) for the Darfield and Christchurch earthquakes (source: Canterbury Geotechnical Database, 2013b).**

[Figure]

**Figure 3 – Location of surface liquefaction observations (brown) in Christchurch and surrounding areas due to the Darfield and Christchurch earthquakes, based on data provided by Tonkin and Taylor and published within the Canterbury Geotechnical Database (CGD 2013a).**

[Figure]

**Figure 4 – Plots comparing observed $V_{S30}$ with $V_{S30}$ estimated from Boore (2004) equations, with respect to observed $V_{S10}$ (left) and observed $V_{S20}$ (right). The dashed lines represent the 95% confidence interval around the Boore (2004) relationships. $V_{S30}$ is the average shear wave velocity in the top 30m of ground and $V_{S10}$ and $V_{S20}$ are the equivalents at 10m and 20m depth respectively.**

[Figure]

**Figure 5 – Receiver Operating Characteristic (ROC) curves for  forecasting models. Refer to Table 4 for descriptions corresponding to model acronyms.**

[Figure]

**Figure 6 – Maps of liquefaction forecasts from selected models for the Darfield earthquake, where the brown areas represent positive liquefaction forecasts, the grey areas represent negative liquefaction forecasts and the white areas are where no forecast was made due to lack of input data. Refer to Table 4 for descriptions corresponding to model acronyms.**

Formatted Table

[Figure]

**Figure 7 – Maps of liquefaction forecasts from selected models for the Christchurch earthquake, where the brown areas represent positive liquefaction forecasts, the grey areas represent negative liquefaction forecasts and the white areas are where no forecast was made due to lack of input data. Refer to Table 4 for descriptions corresponding to model acronyms.**

[Figure]

**Figure 8 – Plots of liquefaction probability against Liquefaction Potential Index (LPI) derived from site specific observations by generalized linear model with probit link function for two best performing LPI models. Plots also display the observed liquefaction rates at each LPI value and classified by sample size**

**Table 1 – Reference list of acronyms used in this paper**

| Acronym | Description |
| --- | --- |
| AOC | Area under ROC curve |
| CGD | Canterbury Geotechnical Database |
| CPT | Cone penetration test |
| CRR | Cyclic resistance ratio |
| CSR | Cyclic stress ratio |
| CTI | Compound topographic index |
| EQC | Earthquake Commission |
| FN | False negative model  forecasts (no.) |
| FP | False positive model  forecasts (no.) |
| FPR | False positive rate ( = FP / Observed negatives) |
| FS | Factor of safety against liquefaction |
| LPI | Liquefaction potential index |
| MCC | Matthew's correlation coefficient |
| MSF | Magnitude scaling factor |
| MWF | Magnitude weighting factor |
| NEHRP | National Earthquake Hazards Reduction Program |
| PGA | Peak ground acceleration |
| PGDf | Permanent ground deformation |
| PGDf$_H$ | Horizontal permanent ground deformation |
| PGDf$_V$ | Vertical permanent ground deformation |
| RMSE | Root mean square error |
| ROC | Receiver operating characteristic |
| SPT | Standard penetration test |
| TN | True negative model  forecasts (no.) |
| TNR | True negative rate ( = TN / Observed negatives) |
| TP | True positive model  forecasts (no.) |
| TPR | True positive rate ( = TP / Observed positives) |
| USGS | United States Geological Survey |

**Table 2 – Reference list of variables used in this paper**

| Variable | Description | Units | Input variables |
|---|---|---|---|
| $a_{max}$ | Peak horizontal ground acceleration | m/s$^2$ | - |
| $CRR$ | Cyclic resistance ratio | - | $MSF$, $V_{S1}$, $V_{S1}^*$ |
| $CSR$ | Cyclic stress ratio | - | $a_{max}$, $\sigma_v$, $\sigma_v'$, $r_d$ |
| $CTI$ | Compound topographic index | - | - |
| $d_w$ | Depth to groundwater | m | - |
| $FS$ | Factor of safety against liquefaction | - | $CRR$, $CSR$ |
| $K_M$ | HAZUS moment magnitude correction factor | - | $M_W$ |
| $K_W$ | HAZUS ground water correction factor | - | $d_w$ |
| $K_\Delta$ | Displacement correction factor | - | $M_W$ |
| $LPI$ | Liquefaction potential index | - | $FS$, $z$ |
| $MSF$ | Magnitude scaling factor | - | $M_W$ |
| $M_W$ | Moment magnitude | - | - |
| $MWF$ | Magnitude weighting factor | - | $M_W$ |
| $ND$ | Normalised distance to coast (Zhu et al., 2015) | - | - |
| $PGA$ | Peak horizontal ground acceleration (non USGS) | g | - |
| $PGA_{M,SM}$ | Peak horizontal ground acceleration (from USGS ShakeMap) | g | - |
| $PGD \mid (PGA / PLSC)$ | HAZUS expected $PGD_f$ for a given liquefaction susceptibility zone | m | $PGA$, liquefaction susceptibility |
| $PGDf$ | Permanent ground deformation | m | - |
| $PGDf_H$ | Horizontal permanent ground deformation | m | - |
| $PGDf_V$ | Vertical permanent ground deformation | m | - |
| $P_{ml}$ | HAZUS proportion of map unit susceptible for a given liquefaction susceptibility zone | - | Liquefaction susceptibility |
| $r_d$ | Shear stress reduction coefficient | - | $z$ |
| $R_f$ | Horizontal distance to surface projection of fault | km | - |

Formatted Table

| Symbol | Description | Unit | Depends on |
|---|---|---|---|
| | rupture | | |
| $T_d$ | Duration between first and last occurrence of $PGA \geq 0.05\mathrm{g}$ | s | - |
| $V_S$ | Shear wave velocity | m/s | - |
| $V_{S1}$ | Stress-corrected shear wave velocity | m/s | $V_S, \sigma_v'$ |
| $V^*_{S1}$ | Limiting upper value of $V_{S1}$ for cyclic liquefaction occurrence | m/s | Fines content |
| $V_{S(0-10)}$ | Average shear wave velocity in top 10m | m/s | - |
| $V_{S(10-20)}$ | Average shear wave velocity between 10m and 20m | m/s | - |
| $V_{S30}$ | Average shear wave velocity in top 30m | m/s | - |
| $z$ | Depth | m | - |
| $\sigma_v$ | Total overburden stress | kPa | $z$, soil density |
| $\sigma_v'$ | Effective overburden stress | kPa | $\sigma_v, z$ |

**Table 3 – Conversion between Canterbury and HAZUS liquefaction susceptibility zones for three implementations of HAZUS method. Refer to Table 4 for descriptions corresponding to model acronyms.**

| Canterbury liquefaction susceptibility zones | Equivalent HAZUS liquefaction susceptibility zone and expected settlement amplitude | | | | | |
|---|---|---|---|---|---|---|
| | Model HAZ1 | | Model HAZ2 | | Model HAZ3 | |
| | Zone | Expected settlement (cm) | Zone | Expected settlement (cm) | Zone | Expected settlement (cm) |
| None | None | 0 | None | 0 | None | 0 |
| Low | Low | 2.5 | Very low | 0 | Average low and very low | 1.25 |
| Moderate | Moderate | 5 | Moderate | 5 | Moderate | 5 |
| High | High | 15 | Very high | 30 | Average high and very high | 22.5 |

**Table 4 – Liquefaction  forecasting models  compared in this paper**

| Model | Description |
|-------|-------------|
| LPI1 | Liquefaction potential index (LPI) with known shear wave velocity, $V_S$, profiles |
| LPI2 | LPI with average shear wave velocity in the top 30m, $V_{S30}$, as a proxy for $V_S$  |
| LPI3 | LPI with simulated $V_S$ profiles |
| LPIref | LPI calculated from standard penetration test (SPT) results |
| HAZ1 | HAZUS with 'direct' conversion of susceptibility zones |
| HAZ2 | HAZUS with 'extreme' susceptibility zones |
| HAZ3 | HAZUS with 'average' conversion of susceptibility |
| ZHU1 | Global model by Zhu et al. (2015) |
| ZHU2 | Regional model by Zhu et al. (2015) |
| ZHU3 | Local model by Zhu et al. (2015) |

Formatted Table

Formatted Table

**Table 5 – Summary of contingency table data and diagnostic scores for all models using initial threshold estimates, including 'LPI' models subject to sensitivity test without Juang et al. (2005) correction factors being applied to the factor of safety. Refer to Table 4 for descriptions corresponding to model acronyms.**

| Model | True Positives ($TP$) | True Negatives ($TN$) | False Positives ($FP$) | False Negatives ($FN$) | True Positive Rate ($TPR$) | True Negative Rate ($TNR$) | False Positive Rate ($FPR$) |
|---|---|---|---|---|---|---|---|
| LPI1 | 6345 | 25685 | 9442 | 1478 | 0.811 | 0.731 | 0.269 |
| LPI2 | 147 | 35063 | 64 | 7676 | 0.019 | 0.998 | 0.002 |
| LPI3 | 37374287 | 3198230578 | 31454549 | 40863536 | 0.478548 | 0.910870 | 0.0900.130 |
| LPIref | 5964 | 20826 | 14301 | 1859 | 0.762 | 0.593 | 0.407 |
| HAZ1 | 0 | 35127 | 0 | 7823 | 0.000 | 1.000 | 0.000 |
| HAZ2 | 0 | 35127 | 0 | 7823 | 0.000 | 1.000 | 0.000 |
| HAZ3 | 0 | 35127 | 0 | 7823 | 0.000 | 1.000 | 0.000 |
| ZHU1 | 1880 | 33483 | 1644 | 5943 | 0.240 | 0.953 | 0.047 |
| ZHU2 | 3135 | 31931 | 3196 | 4688 | 0.401 | 0.909 | 0.091 |
| ZHU3 | 2754 | 31017 | 4110 | 5069 | 0.352 | 0.883 | 0.117 |
| LPI2b | 610 | 34902 | 225 | 7213 | 0.078 | 0.994 | 0.006 |
| LPI3b | 6068 | 20509 | 14618 | 1755 | 0.806 | 0.584 | 0.416 |

**Table 6 – Summary of contingency table data and diagnostic scores for LPI models subject to sensitivity test without Juang et al. (2005) correction factors being applied to *FS***

| Model | *TP* | *TN* | *FP* | *FN* | *TPR* | *TNR* | *FPR* |
|---|---|---|---|---|---|---|---|
| LPI2b | 610 | 34902 | 225 | 7213 | 0.078 | 0.994 | 0.006 |
| LPI3b | 5002 | 27136 | 7991 | 2821 | 0.639 | 0.773 | 0.227 |

Table 6 – Model quality diagnostics and optimum threshold values for each model from ROC curves. Refer to Table 4 for descriptions corresponding to model acronyms.

| Model | Area Under Curve (*AUC*) | *J*-statistic | Threshold | True Positive Rate (*TPR*) | True Negative Rate (*TNR*) |
|---|---|---|---|---|---|
| LPI1 | 0.845 | 0.573 | 7 | 0.774 | 0.799 |
| LPI2 | 0.630 | 0.122 | 1 | 0.131 | 0.991 |
| LPI2b | 0.630 | 0.206 | 1 | 0.224 | 0.982 |
| LPI3 | 0.76272 | 0.41420 | 14 | 0.671581 | 0.742839 |
| LPI3b | 0.76166 | 0.4154 | 410 | 0.64617 | 0.76997 |
| LPIref | 0.748 | 0.366 | 6 | 0.689 | 0.678 |
| HAZ1 | 0.679 | 0.238 | 0.1 | 0.073 | 0.999 |
| HAZ2 | 0.608 | 0.316 | 0.1 | 0.134 | 0.997 |
| HAZ3 | 0.661 | 0.315 | 0.1 | 0.133 | 0.998 |
| ZHU1 | 0.753 | 0.355 | 0.1 | 0.556 | 0.799 |
| ZHU2 | 0.760 | 0.371 | 0.1 | 0.767 | 0.604 |
| ZHU3 | 0.718 | 0.306 | 0.1 | 0.712 | 0.594 |

5 Table 7 – Coefficients of link function and summary of contingency table analysis for the two best performing LPI models. Refer to Table 4 for descriptions corresponding to model acronyms.

| Model | $\beta_1$ | $B_0$ | True Positive Rate (*TPR*) | True Negative Rate (*TNR*) | *J*-statistic | Area Under Curve (*AUC*) |
|---|---|---|---|---|---|---|
| LPI1 | 0.067 | -1.555 | 0.683 | 0.869 | 0.551 | 0.843 |
| LPI3b | 0.08298 | -1.376299 | 0.70484 | 0.86056 | 0.564641 | 0.7606 |

Table 9 – Expected settlement amplitudes for liquefaction susceptibility zones from HAZUS methodology (NIBS, 2003)

| Liquefaction susceptibility zone | Expected settlement amplitude (inches) |
|---|---|
| Very high | 12 |
| High | 6 |

| | |
|---|---|
| Moderate | 2 |
| Low | 1 |
| Very low | 0 |
| None | 0 |

**Table 8 – Summary statistics of vertical ground deformation (PGDf$_V$) predictions estimates for Darfield and Christchurch earthquakes from HAZUS models. Refer to Table 4 for descriptions corresponding to model acronyms.**

| Score | Observed | Vertical permanent ground deformation, PGDf$_V$ (m) | | |
|---|---|---|---|---|
| | | HAZ1 | HAZ2 | HAZ3 |
| Pearson R$^2$ | n/a | 0.064 | 0.051 | 0.058 |
| Mean | 0.118 | 0.003 | 0.008 | 0.005 |
| Minimum | 0.000 | 0.000 | 0.000 | 0.000 |
| Lower quartile | 0.051 | 0.000 | 0.000 | 0.000 |
| Median | 0.100 | 0.001 | 0.000 | 0.000 |
| Upper quartile | 0.162 | 0.004 | 0.004 | 0.004 |
| Maximum | 1.464 | 0.022 | 0.066 | 0.043 |
| Residual mean | n/a | -0.114 | -0.110 | -0.112 |
| Root-mean-square errorRMSE | n/a | 0.146 | 0.142 | 0.144 |

**Table 9 – Summary statistics of horizontal permanent ground deformation (PGDf$_H$) predictions estimates for Darfield and Christchurch earthquakes from EPOLLS and HAZUS models. Refer to Table 4 for descriptions corresponding to model acronyms.**

| Score |  | Horizontal permanent ground deformation, PGDf$_H$ (m) | | | | |
|---|---|---|---|---|---|---|
| | Observed | EPOLLS | HAZ1 | HAZ2 | HAZ3 |
| Pearson R$^2$ | n/a | 0.000 | 0.022 | 0.032 | 0.027 |
| Mean | 0.269 | 0.682 | 0.141 | 0.172 | 0.150 |
| Minimum | 0.001 | 0.149 | 0.000 | 0.000 | 0.000 |
| Lower quartile | 0.124 | 0.418 | 0.000 | 0.000 | 0.000 |

| | | | | | |
|---|---|---|---|---|---|
| Median | 0.206 | 0.748 | 0.084 | 0.050 | 0.067 |
| Upper quartile | 0.312 | 0.964 | 0.184 | 0.191 | 0.182 |
| Maximum | 3.856 | 1.989 | 1.872 | 3.205 | 2.443 |
| Residual mean | n/a | 0.413 | -0.128 | -0.096 | -0.118 |
| Root-mean-square errorRMSE | n/a | 0.582 | 0.345 | 0.438 | 0.376 |